# A data-consistent model of the last glaciation in the Alps achieved with physics-driven AI

Tancrède P. M. Leger [1,2,6] ✉, Guillaume Jouvet [1,6], Sarah Kamleitner [1,3], Jürgen Mey [4], Frédéric Herman [1], Brandon D. Finley [1], Susan Ivy-Ochs[5], Andreas Vieli [3], Andreas Henz [3] & Samuel U. Nussbaumer [3]

25 thousand years ago, the European Alps were covered by the kilometre-thick Alpine Ice Field. Numerical modelling of this glaciation has been challenged by model-data disagreements, including overestimations of ice thickness. We tackle this issue by applying the Instructed Glacier Model, a three-dimensional model enhanced with physics-informed machine learning. This approach allows us to produce 100 Alps-wide and 17 thousand-year-long simulations at 300 m resolution. Previously unfeasible due to computational costs, our experiment both increases model-data agreement in ice extent and reduces the offset in ice thickness by between 200% and 450% relative to previous studies. Our results have implications for better estimating former ice velocities, ice temperature, basal conditions, erosion processes, and paleoclimate in the Alps. This study demonstrates that physics-informed machine learning can help overcome the bottleneck of high-resolution glacier modelling and better test parameterisations, both of which are required to accurately describe complex topographies and ice dynamics.

The extent, thickness, and flow geometry of the European Alpine Ice Field (AIF) during Quaternary glaciations has been studied for more than a century[1-4]. The result is an abundant library of mapped and dated ice-contact glaciogenic landforms and sediments (e.g. moraines and tills)[5-8], glaciofluvial deposits[9], and trimlines[10-16], making the extent of the AIF during the Last Glacial Maximum (LGM: ~30–19 ka[17,18]) one of the most well known in the world[11,18]. Yet, numerous controversies remain regarding the spatio-temporal evolution of the AIF's former thickness, velocity patterns, basal conditions, outlet glacier asynchronies, erosion potential, and isostatic adjustment. Obtaining model simulations of the LGM ice field that match empirical data has implications for understanding such mechanisms and can also improve reconstructions of late-Quaternary human and ecological history[19].

Under these motivations, over the past 20 years, several studies[20-24] have produced simulations of AIF glaciations using glacier evolution models such as the Parallel Ice Sheet Model (PISM[25]) and the integrated second-order shallow ice approximation model (iSOSIA[26]), with the main objective to closely fit moraine and trimline evidence. As a result, model-data misfits in LGM ice extent were incrementally reduced, with the latest study[22] making progress by using an improved climate forcing[27]. However, a major discrepancy has persisted when comparing model results with trimlines assumed to indicate past maximum ice surface elevations[10-16]. Numerical models have traditionally overestimated the AIF's thickness by between 400–1000 metres during the local LGM[20,22], thus misrepresenting the supposedly thin and topographically constrained nature of the ice field[11,18]. This ice thickness overestimation has been attributed to oversimplified ice flow dynamics and the coarse spatial resolutions of previous AIF-wide models, computationally restricted to 1–2 km[22]. However, the discrepancy also caused some studies[20,28,29] to question whether trimlines in the Alps are true indicators of maximum ice surface elevations, or rather of warm-to-cold-based ice transitions, as was previously

[1]Institute of Earth Surface Dynamics, University of Lausanne, Lausanne, Switzerland. [2]Department of Geography, University of Sheffield, Sheffield, UK. [3]Department of Geography, University of Zurich, Zurich, Switzerland. [4]Institute of Environmental Science and Geography, University of Potsdam, Potsdam, Germany. [5]Laboratory of Ion Beam Physics, ETH Zurich, Zurich, Switzerland. [6]These authors contributed equally: Tancrède P. M. Leger, Guillaume Jouvet. ✉e-mail: tancrede.leger@unil.ch

debated in other glaciated regions, such as Britain and Scandinavia for instance[30].

To address this conundrum, we apply a different approach by modelling the AIF with the Instructed Glacier Model (IGM)[31,32], a thermo-mechanically coupled three-dimensional glacier evolution model. IGM makes use of recent improvements in physics-informed machine learning to accelerate the high-order Blatter-Pattyn solver of ice flow[33], enabling efficient computation on graphics processing units (GPU). The resulting model simulates glacier motion with fidelity while offering the same ability to perturbate parameters controlling ice properties as traditional central-processing-unit-based models, but at a fraction of the computational cost. Here, we use IGM to simulate the AIF's transient evolution from 35 to 18 ka, thus bracketing the full LGM period, at a spatial resolution of 300 m, an order of magnitude higher than previously achieved[20]. Computationally unfeasible with traditional glacier models, our perturbed parameter ensemble experiment shows that increasing spatial resolution substantially reduces model-data misfits in both LGM ice extent and thickness across the Alps (Fig. 1), resulting in a thinner ice field than previous 1–2 km resolution models[20,22]. This approach enables us to produce a high-resolution, data-consistent, yet physics-based transient simulation of the AIF's last glaciation. The results have wider implications for addressing open research questions on Quaternary glacial erosion processes, post-glacial isostatic rebound, and paleoclimate in the European Alps. Our study also reveals that GPU-based glacier evolution modelling is a promising tool for better reconstructing other paleo ice fields, but also for past and future ice-sheet-scale modelling. Indeed, by substantially reducing computational costs, this approach permits advances in modelling resolution and parameter space exploration, both essential to resolve complex topographies and model more accurate past and future glaciated environments.

## Results

### High resolution model setup

To model the transient evolution of the AIF during the LGM, we design a high-resolution (300 m) model setup with IGM[31,32] that builds on the Parallel Ice Sheet Model (PISM[25]) experimental setup of Jouvet et al.[22]. We implement an equivalent enthalpy module formulated by Aschwanden et al.[34] for modelling polythermal ice (see 'Methods' section) and use the same input climate fields[27] and geothermal heat flux data[35]. Surface mass balance is computed using a comparable positive degree-day scheme based on Calov & Greve[36]. We use a pseudo-plastic power slip law[37] and parameterize the space-dependent yield stress to be controlled by the effective pressure and frictional strength of basal till[38] (see 'Methods' section). Glacial isostasy is modelled by coupling IGM with the gFlex lithospheric flexure model[39], after Mey et al.[21].

The experiment setup of Jouvet et al.[22] is however improved by using an Alps-specific climate record (instead of the EPICA[40] ice core record) as input to our glacial index scheme[22,41] (Fig. 2e). The applied signal combines the Bergsee lacustrine record[42] (35–30 ka: Black Forest, Southern Germany) and the Sieben Hengste speleothem $\delta^{18}O$ record[43] (30–18 ka: Bernese Alps, Switzerland). Further improvements of the setup include an avalanche scheme re-distributing snow and ice accumulated on steep slopes (>45°) down to the glacier surface, and an elevation-dependent parameterization of the basal till friction angle, enabling a spatially-variable bed strength (see 'Methods' section). In the main overdeepened valleys of the Alps (e.g. the Rhône valley), the thickness of unconsolidated infill sediments present before the last glaciation remains debated[21,44]. Here, we test the impact of valley-fill sediments on the ice field's LGM geometry by using two distinct digital elevation models within our ensemble (see Supplementary Table 1, 'Methods' section). In the first digital elevation model (from Jouvet et al.[22]), only present-day lakes and glaciers are removed[45], while the second (from Mey et al.[21]) additionally removes all valley-fill sediments from major valleys across the Alps.

In this study, IGM makes use of a physics-informed convolutional neural network to emulate a high-order (three-dimensional) Blatter-Pattyn ice-flow solver[32] (see 'Methods' section). This methodology reduces computational costs by several orders of magnitude through efficient parallelization on GPU. This approach enables us to run AIF-wide simulations (542,700 km²) at a spatial resolution of 300 m (grid size: 3006 x 2006). Over the entire Alps, a 300 m grid increases the total cell number for which to compute ice flow by a factor of ~11 relative to 1 km (best achieved so far[20]). While running a 20 kyr-long simulation at 300 m would require ~2.5 years with PISM using a 32-core (3.7 GHz) computer (or ~6 months using a state-of-the-art high-performance computer), it is achieved in only ~2.5 days with IGM using a single GPU (Nvidia RTX 4090).

Taking advantage of such capabilities, and after validating IGM's suitability through quantitative comparison with earlier PISM results (see 'Methods' section), we ran AIF-wide simulations covering the full LGM period, from 35 to 18 ka. To assess model-data agreement (see below) and investigate the model sensitivity to the choice of parameters for a wide range of ice field evolution scenarios, we perform 100 simulations. Within this ensemble, we vary 10 key model parameters drawn from eight different components of the glacier system which modify both ice properties and boundary conditions (see Supplementary Table 1). The 100 parameter sets are sampled between given parameter ranges using a Latin hypercube algorithm[46], optimised with a maximin criterion. This commonly used technique provides efficient and homogeneous sampling of parameter spaces with high dimensionality, helping to optimally explore the variety of possible model responses[46].

### Model-data comparison scheme

Model-data agreement is first evaluated for ice extent by comparing modelled maximum ice thickness fields against an empirical reconstruction of AIF margins at the LGM originally produced by Ehlers et al.[17]. In this study, we use the newest version of this outline (Fig. 1a), which was updated by a series of studies[5,8,47–53] providing more constraints on LGM margins of main AIF outlet glaciers (see 'Methods' section), and which is used widely within the community[5,6,8,18,52,54]. The fit for each ensemble simulation is quantitatively assessed by computing to total number of overlapping pixels between the modelled LGM ice margin and rasterized polygons mapped around the empirical LGM outline (see Figs. 1a, 2b, Supplementary Fig. 19). For this test, LGM moraine polygons are mapped exclusively in regions of high confidence in the position and broad LGM timing (35–18 kyr BP) of the AIF margin due to abundant mapping and dating[5,6,18]. The overlapping pixel counts are subsequently normalized to obtain a score between 0 (worse-fit simulation) and 1 (best-fit simulation). As a first model-data comparison sieve, any simulations scoring below 0.8 ($n = 58$) at this test were excluded (Fig. 2b). Here, the modelled LGM state for each simulation is obtained by computing the maximum thickness reached in each pixel at any time during the simulation, thus providing a time-independent map of maximum modelled thickness and extent (Fig. 1a). The precise dated timing of the LGM in specific regions is thus not directly compared with the timing of the modelled LGM as part of this model-data comparison scheme, which exclusively aims at finding simulations with the most data-consistent AIF geometry (see Supplementary Fig. 19).

Secondly, we assess model-data fit in ice thickness for the remaining simulations by computing the difference between modelled maximum ice surface elevations and 353 reported trimline elevations from the literature[10–16] (Figs. 1b, 2d, see 'Methods' section). To ensure a direct comparison, modelled surface elevations are corrected by computing and adding the LGM lithospheric deflection at each trimline location. Finding simulations with lowest mean and standard deviation values in elevation difference helped minimise both the

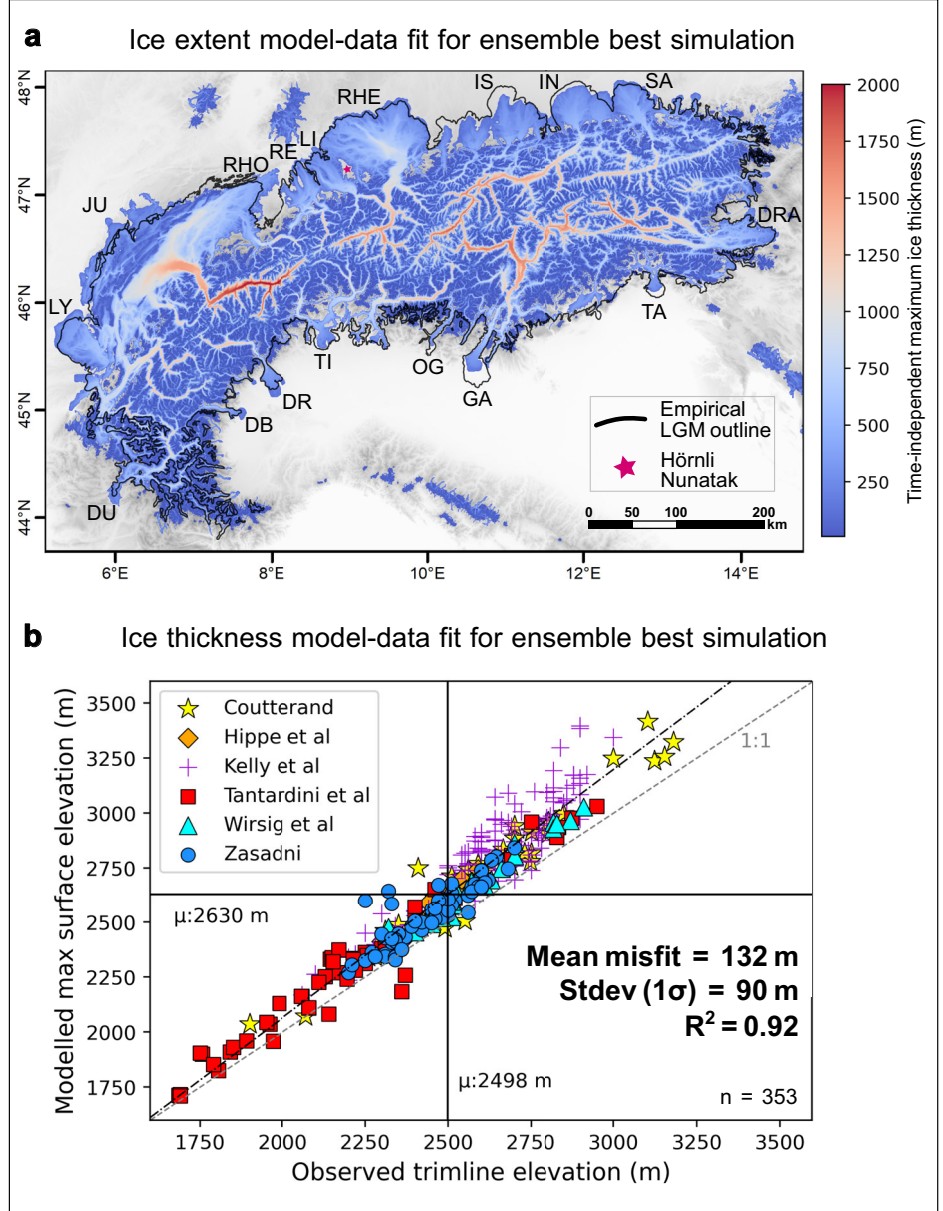

**Fig. 1 | Best-fit Instructed Glacier Model (IGM) 300 m simulation (number 37) of the Alpine Ice Field during the Last Glacial Maximum (LGM). a** Shows the 2-D field of modelled time-independent maximum ice thickness for simulation 37, enabling the comparison between modelled LGM ice extent and the empirical LGM outline of the Alpine Ice Field used in this study (black line, modified from Ehlers et al.[17]). Black acronyms stand for main outlet glacier names: i.e. the Rhône (RHO), Jura (JU), Lyon (LY), Durance (DU), Dora-Baltea (DB), Dora-Riparia (DR), Oglio (OG), Tagliamento (TA), Drau (DRA), Salzach (SA), Inn (IN), Isar (IS), Rhein (RHE), Linth (LI), Reuss (RE) outlet glaciers. The pink star highlights the location of the Hörnli nunatak. Note that simulation 37 was modelled using a basal topography with valley fill sediments (See Supplementary Figs. 5, 6). **b** Displays the model-data

agreement in Alpine Ice Field ice thickness shown for our best-fit IGM ensemble simulation (number 37), obtained by computing the difference between 353 quality-controlled trimline elevations from literature[10–16] (see Methods section: 'Empirical trimline elevation data') and maximum modelled ice surface elevations (time-independent). The black dash-dotted line represents the best-fit linear regression for the data ($R^2 = 0.92$). $R^2$ stands for coefficient of determination. Note that the mean and standard deviation (Stdev) of misfit displayed on panel b are exclusive results of simulation 37, while statistics discussed in main text and reported as the official study results (+ 146 ± 12 m) are averaged over our eight Not-Ruled-Out-Yet simulations (NROYs): see 'Results' section.

overall model-data misfit in ice thickness and its scatter, the latter acting as a proxy for spatial variability in model-data agreement (Fig. 2d, f). Simulations with mean and standard deviation values exceeding 160 m and 100 m in elevation difference, respectively, were sieved out ($n = 32$). This quantitative three-sieve model-data comparison procedure was designed to isolate a group of 10 best-fit simulations (10% of the ensemble) for further inspection. Finally, a visual check of these 10 simulations by three experts enabled to identify two outlier simulations which produce far too extensive ice margins

throughout the Western and Northwestern Alps. After removing these two outliers, the final eight Not-Ruled-Out-Yet simulations (NROYs) are found to be indistinguishable in quality of ice-extent fit with the empirical LGM outline and are thus used for all subsequent quantitative results reported in this study, using NROY means and standard deviations in given metrics (Fig. 2). For visualisation and discussion purposes only, simulation 37 was further selected as our best-fit simulation, due to presenting the best combination of extent and thickness model-data fit.

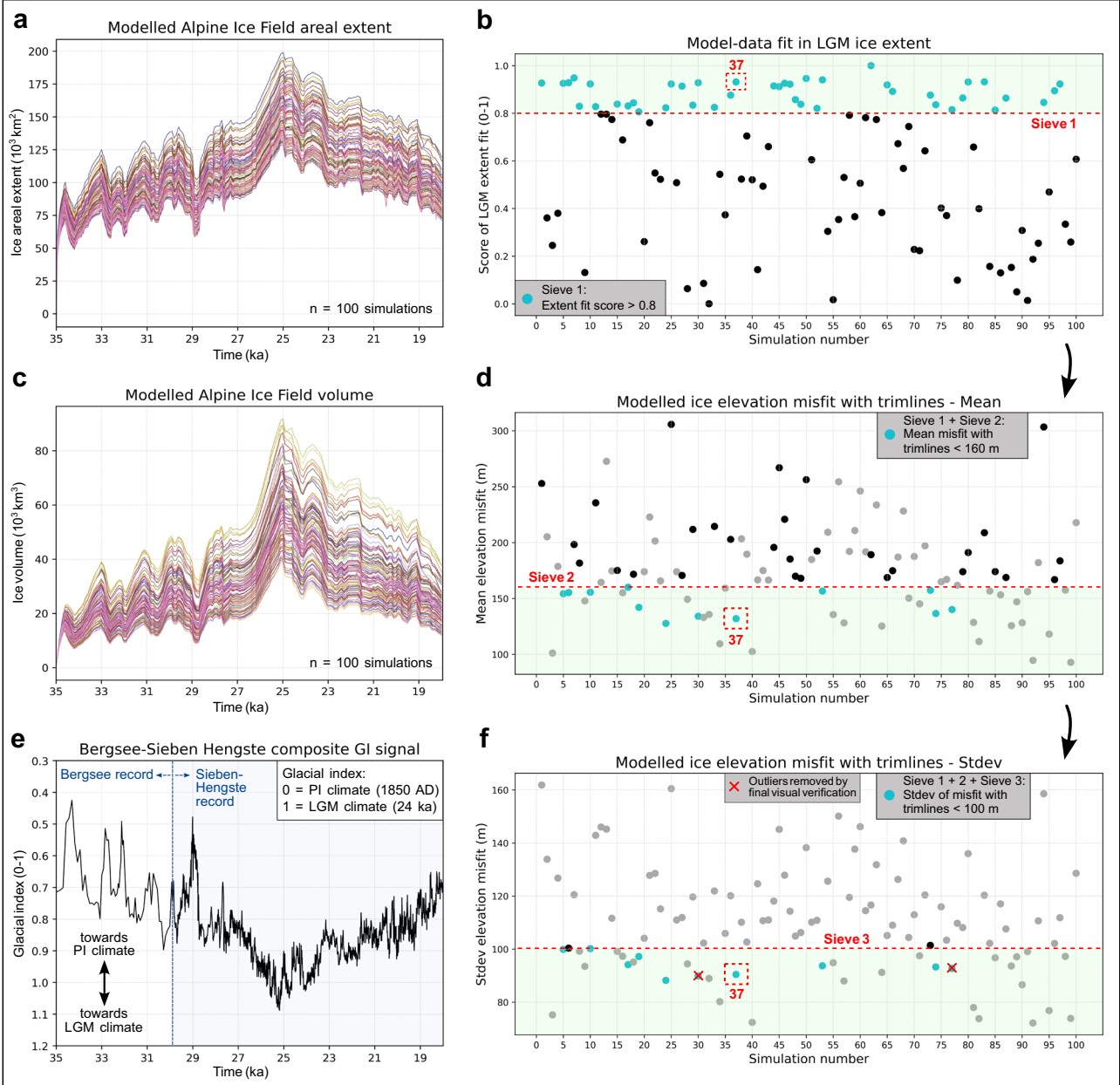

**Fig. 2 | Instructed Glacier Model (IGM) 300 m ensemble results ($n$ = 100 transient simulations). a, c** Indicate the modelled Alpine Ice Field areal extent and volume, fluctuating in line with the combined climate signal[42,43] used as input to our glacial index scheme (**e**). **b**, **d** and **f** Show the model-data comparison scheme applied to isolate best-fit simulations (NROYs: light blue dots in panel f) with a 3-sieves approach. For each of panels **b**, **d**, and **f**, grey dots are simulations removed by previous sieves while black dots are simulations removed by current sieve (e.g.

black dots in panel f are simulations removed exclusively by sieve 3). The two red crosses in panel f represent two outlier simulations identified through final visual check and removed from the final pool of Not Ruled-Out Yet simulations (NROYs). Final NROYs (simulations 5, 10, 17, 19, 24, 37, 53, and 74) are indistinguishable in quality of model-data ice extent fit with the Last Glacial Maximum (LGM) empirical outline used in this study. Stdev stands for standard deviation, PI for Pre-Industrial, and GI for Glacial Index.

## Improved LGM model-data fit

Overall, our model produces a thinner LGM ice field resulting in more valley-confined ice flow (Fig. 3), and leading to a substantial increase in model-data fit in ice surface elevation. Using the same trimline dataset[10] as Jouvet et al.[22] and Seguinot et al.[20] ($n$ = 175), the mean model-data misfit with observed trimlines for our best-fit IGM simulations is +191 ± 15 m (NROY mean). The positive bias in modelled ice thickness is reduced by 200% and 450% compared with these earlier studies, respectively. When using a more extensive ($n$ = 353), spatially widespread, and quality-controlled trimline dataset[10,11,13–16] (Figs. 1, 2, 4b, see 'Methods' section), the mean misfit value is further reduced to

+146 ± 12 m (NROY mean). Across the Alps, modelled maximum ice surface elevations are positively and well correlated with reported trimline elevations (NROY mean $r^2$ = 0.92) (see Supplementary Figs. 2, 3), with misfit values for all trimlines presenting limited scatter (NROY-mean stdev=94 ± 4 m). Despite varying 10 key parameters conservatively (see Supplementary Table 1), all 100 simulations present mean trimline misfit values that remain below previous and coarser-resolution models' best estimates[20,22]. This implies that increasing model resolution from 2 km to 300 m systematically produces a thinner LGM AIF throughout the Alps (Fig. 3a), on average by 218 ± 15 m (NROY mean) relative to Jouvet et al.[22]. Indeed, maximum AIF volume

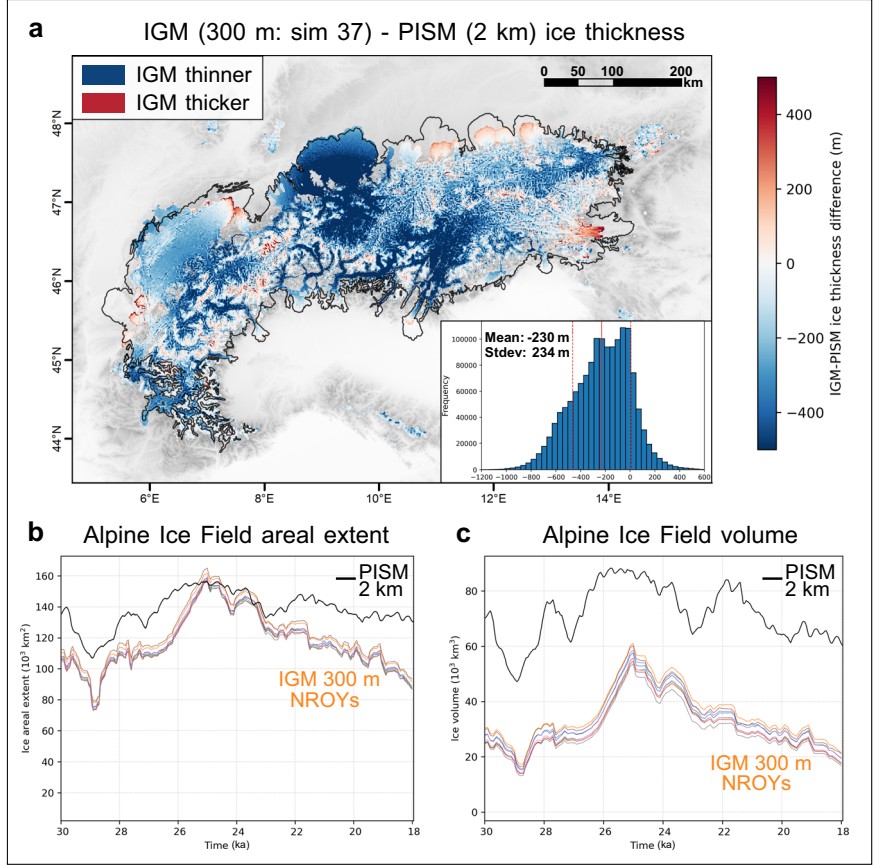

**Fig. 3 | Difference in modelled Last Glacial Maximum (LGM) ice thickness between 300 m and 2 km.** Here, our best-fit Instructed Glacier Model (IGM) simulation with a spatial resolution of 300 m (number 37) is compared against the Parallel Ice Sheet Model (PISM) simulation at 2 km resolution by Jouvet et al.[22] (panel **a**). Stdev stands for standard deviation. The empirical LGM outline of the Alpine Ice Field used in this study (modified from Ehlers et al.[17]) is highlighted by the black line. **b**, **c** present time series (30–18 ka) of modelled Alpine Ice Field areal extent and volume for both our IGM 300 m Not-Ruled-Out-Yet simulations (NROYs, *n* = 8, coloured lines) and the 2 km PISM simulation[22]. Note panel a only displays coloured pixels where ice is modelled by both IGM and PISM simulations (when maximum ice thickness is reached in each pixel), as this is required to compute the thickness difference. The full LGM ice extent modelled by our best-fit simulation is instead shown in Fig. 1a.

$(57.6 \pm 2.1 \times 10^3 \text{ km}^3)$ is here ~35% lower than when modelled at 2 km $(88.3 \times 10^3 \text{ km}^3)$[22], while maximum AIF area is <2% different (Fig. 3). The resulting ice thickness difference is spatially heterogeneous, however, and most prominent for the Rhein and Garda outlet glacier catchments. The NROYs, for instance, model both the adjacent Rhein and Linth outlet glaciers reaching the mapped LGM moraines (Figs. 1, 3) while keeping the Hörnli nunatak (47.4°N, 8.95°E) and its surrounding hills ice free. Although well documented[7,55], this LGM ice-field configuration was never achieved by previous AIF-wide simulations, which instead modelled ice covering the entire region[20,22].

The large and recurring model-data thickness offset obtained in previous modelling efforts caused certain studies[20,28,29] to suggest that trimlines in the Alps may represent a warm-to-cold-based ice transition. Across our 100 simulations, we find highly variable modelled basal ice temperatures at the locations of trimlines during the LGM, with no apparent clusters towards distinct temperatures. This suggests no relationship exists in our modelling between thermal boundaries of basal ice and trimline formation (see Supplementary Fig. 4). These results suggest that in the Alps, observed trimlines are likely true indicators of maximum ice surface elevations during Quaternary glacial maxima.

Motivated by potential uncertainties in the input climate forcing[27] and our dependence to the glacial index approach[41], our ensemble features a catchment-specific precipitation offset scheme (see 'Methods' section). The latter permits simulation-specific climate variations

with possible localised changes in input precipitation of between −40% and +50%. Together with higher-resolution modelling, our approach also improves model-data fit in LGM ice extent throughout the AIF, relative to previous work (Figs. 1a, 4a). Where Jouvet et al.[22]'s PISM simulation produced AIF margins more extensive than empirical reconstructions (i.e. in the Rhein, Reuss, Durance, Jura, Vittorio-Veneto, and Astico regions), our best-fit simulations produce less extensive, more data-consistent ice margins. Conversely, the fit is improved where previously-modelled AIF margins[22] did not reach the mapped ice margins (i.e. in the Drau, Inn, Isar, and Lyon regions) (Fig. 4a). The LGM extent of the Drau outlet glacier (Figs. 1a, 4a), for instance, was notoriously challenging to model accurately[20,22,24]. Our NROYs reproduce its extent within 10 km of well-preserved and mapped local terminal moraines[9], and present a margin shape consistent with the consensus empirical outline[17]. However, we note three instances, i.e. the Rhône, Dora-Baltea and Isère outlet glaciers, where our NROYs model too extensive ice margins leading to a worse fit with data than Jouvet et al.[22]'s model (Fig. 4a). We believe remaining model-data misfits in ice thickness (+146 ± 12 m, NROY mean) and in the extent of certain outlet glaciers can either be, 1) Reduced further with higher-than-300 m resolution simulations and more extensive parameter-space explorations using larger ensembles, 2) Related to unavoidable uncertainties inherent to paleo trimline and LGM moraine identification due to challenges with landform preservation and dating, and 3) Related to uncertainties and biases in our input climate and

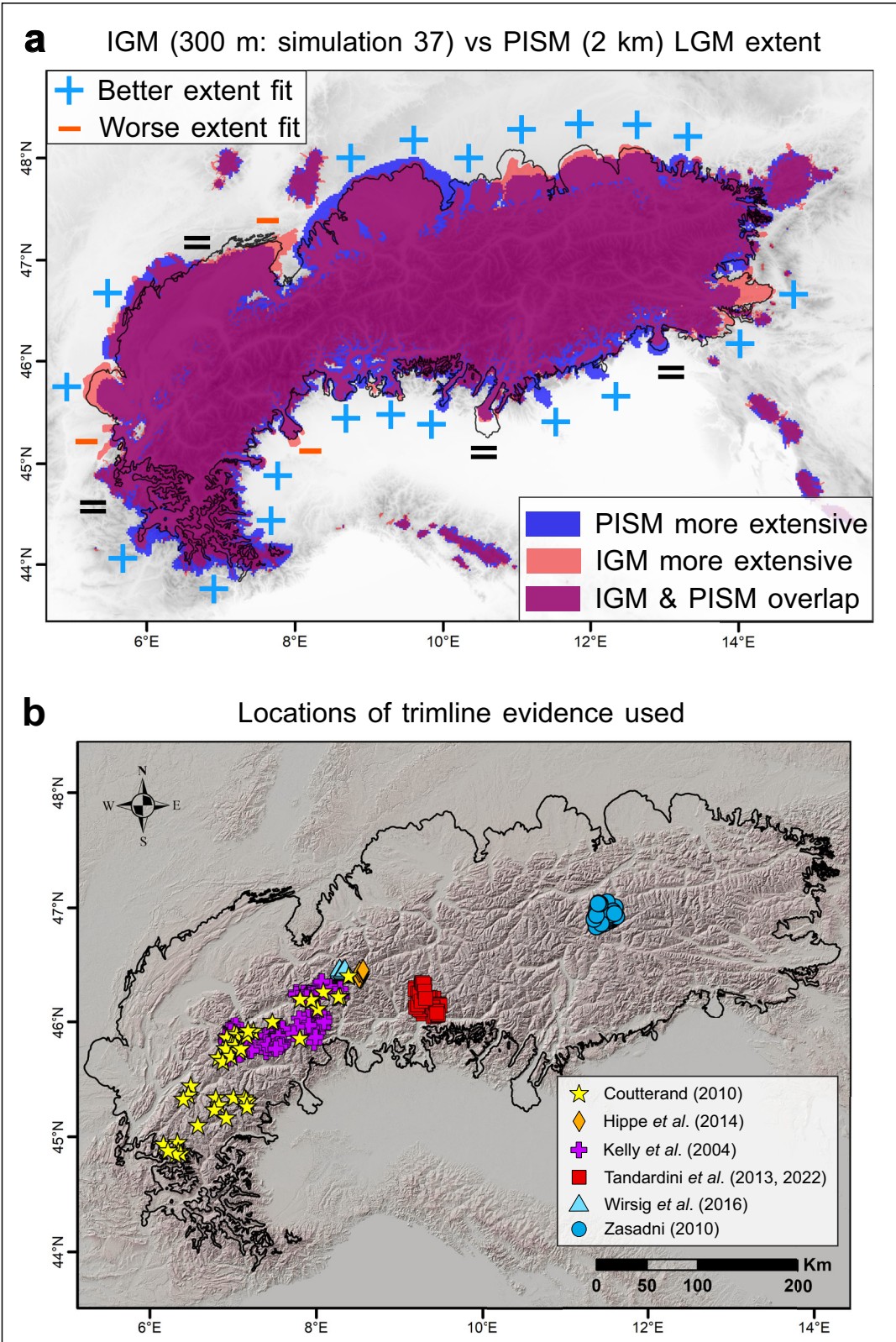

**Fig. 4 | Best-fit Instructed Glacier Model (IGM) 300 m simulation (number 37) results.** **a** Highlights the change in Last Glacial Maximum (LGM) ice extent fit between our best-fit IGM simulation (simulation 37) and the 2 km Parallel Ice Sheet Model (PISM) simulation by Jouvet et al.[22]. The equal symbols suggest model-data fit in ice extent is comparable between IGM at 300 m and PISM at 2 km. Panel b highlights the locations of the 353 quality-controlled trimline elevations from literature[10-16] (see 'Methods' section), as well as the empirical LGM outline of the Alpine Ice Field used in this study (modified from Ehlers et al.[17]) in black. The map in panel b was produced using ArcGIS Pro 3.2.2 (Esri), and displays the 30 m digital elevation model from ALOS World 3D DEM[82] (version 4.1).

surface mass balance parameterisation. A combination of impact from these three mechanisms is also considered likely. For the latter, future improvements in the input climate through higher-resolution regional climate modelling and more transient simulations accompanied by a more complex energy balance model may help reduce some of the remaining model-data misfits.

### Intra-NROY variability

Overall, our NROYs show few clusters within the sampled ranges of the 10 ensemble-varying parameters (see Supplementary Fig. 5). An exception is the Glen's flow law enhancement factor (E), with NROY values all greater than one. Thus, good model-data fit may require more deformable ice than modelled with default flow law constants ($E = 1$, $n = 3$). The lack of clusters in other parameters indicates a better model-data fit can be obtained with highly variable parameter values, making optimal parameter configurations difficult to predict. This variability justifies a perturbed ensemble approach to maximise model-data fit, as numerous local minima may exist. Moreover, two out of eight NROYs were achieved using a basal topography without valley-fill sediments, while the remaining six were not (see Supplementary Fig. 5). A sensitivity analysis on the best-fit simulation (37) further reveals that at the AIF scale, the removal of thick (>300 m) valley-fill sediments prior to glacier advance does not significantly impact ice flux, glacier build-up rates, ice-field geometry, and thus model-data fit at the LGM (see Supplementary Fig. 6). Therefore, we find inaccurate thickness estimations of valley-fill deposits prior to the last glaciation have little impact on the quality of AIF-wide models of the LGM.

## Discussion

Our higher-resolution model simulations (300 m instead of 1–2 km in previous studies[20,22]) result in thinner ice and glacier surface lowering across the AIF during the LGM. We hypothesise that localised ice flow speedup is the main underlying mechanism responsible for this result. Indeed, our NROYs show increased ice velocities within main valley troughs and certain outlet glacier tongues relative to previous coarser models[20,22] (Figs. 5c, d, 6a, 7). At these locations, NROYs produce a more topographically constrained ice flux, with ice flowing into narrower, deeper valleys featuring steeper sidewalls and more pronounced topographic bottlenecks (Figs. 5, 6a). When ice in such valleys drains large glacier catchments, NROYs produce greater flow convergence, with higher-magnitude venturi effects causing ice flow speedups (Figs. 5, 7). The venturi effect here applies as it describes the increase in an incompressible fluid's velocity as it passes through a constriction, in order to respect the principle of mass conservation. Moreover, we expect this positive correlation between bed resolution and flow speed in regions of highly topographically constrained ice flux (narrow, deep valleys) to be more pronounced when solving high-order (three-dimensional), rather than zero-order or hybrid (e.g. PISM), ice flow physics. Here, the high-order Blatter-Pattyn solver[33] used in IGM[32] considers both the vertical and horizontal components of strain through the ice column[56], which helps produce strong ice velocity gradients and promotes the formation of lateral shear margins in valley-confined settings, thus encouraging higher glacier velocities when increasing spatial resolution[57].

The largest velocity increases (+ 600 m yr⁻¹) are found within topographic bottlenecks and deep valleys of the Rhône, Rhein, Dora-Baltea, Garda, Inn and Isar glacier catchments (Fig. 7). We also find a greater modelled AIF-mean surface slope of 5.9 ° at 300 m versus 2.0 ° at 2 km resolution[22]. A more efficient drainage configuration associated with steeper ice surfaces necessarily causes ice thinning for a given accumulation rate, due to mass conservation. Therefore, combined with a better resolved topography producing deeper valleys, we argue localised flow speedups explain the resulting lowering of modelled ice surfaces (Fig. 5), leading to an improved fit with observed trimline elevations (Fig. 1b).

When averaged over the entire AIF, however, differences in ice flux between models of 300 m and 2 km[22] resolution remain small (see Supplementary Figs. 7, 8), with a NROY-mean difference in depth-averaged velocity of only +12 ± 6 m yr⁻¹. This is likely associated with the compensating effect of upper accumulation zones, where a 300 m resolution model leads to lower velocities relative to 1–2 km models. Indeed, a higher-resolution topography resolves high-elevation summits exposed to lower air temperatures, thus increasing the potential for cold-based ice. Here, we find modelled cold-based regions cover 14 ± 3% of the AIF (NROY mean) (Fig. 6b), slightly more than the ~11% modelled at 2 km resolution[22]. Due to reduced basal melt and increased effective pressure, regions of increased resistance to basal sliding (yield stress >5 x 10⁵ Pa) thus cover a larger proportion of the AIF. However, the majority of the ice volume is located in deep Alpine valleys draining large catchments, where higher-resolution modelling causes faster flow and thinning. Hence, although regions of slower and faster flow between 300 m and 2 km[22] models cover similar total areas, localised flow speedups force negative thickness differences to dominate. As a result, when modelling at 300 m versus 2 km[22], the LGM volume of the entire AIF decreases substantially (~ 35%) (Fig. 3).

The improved model and representation of faster ice flow in main valleys has implications for better constraining Quaternary Alpine landscape evolution. Indeed, the localised flow speedups modelled with our NROYs could imply higher-than-previously-estimated erosion rates within major valleys and topographic bottlenecks of the Alps during Pleistocene glacial maxima. Using a non-linear erosion law[58], we find that higher localised basal velocities obtained at 300 m resolution increase the subglacial erosion potential by factors of 2–10 relative to a 2 km model[22] (see Supplementary Figs. 7, 9). In our NROYs, basal velocities at the LGM typically reach values of 800–1300 m yr⁻¹ within regions of topographically constrained fast flow (e.g. the lower Garda valley, Fig. 6a). Depending on the chosen erosion law constant and velocity exponent[59,60], and assuming no negative feedback from eroded material shielding bedrock, such basal velocities could result in erosion rate magnitudes of up to tens of centimetres per year. As previously shown[59,61], regions of fast-flowing ice can thus generate highly localized, rapid valley overdeepening. This could help explain the disproportioned incision of major Alpine valleys draining large catchments, such as the Rhône valley towards Martigny (bedrock ~500 m below sea level)[62], argued to have been deepened by 1–1.5 km in about 1 million years[63], for instance. This highlights the importance of high-resolution glacier modelling for accurate quantification of Pleistocene valley incision rates across the European Alps. Key questions remain concerning the transition time from pre-glacial to current topography and the episodic versus periodic nature of Pleistocene incision signals in the Alps[63]. Our high-resolution, more data-consistent simulation and GPU-based approach could help design future experiments to explore such research questions.

Moreover, the improved model-data agreement enables us to reconstruct potential asynchronies in the waxing and waning signals of different AIF outlet glaciers during the LGM. NROY-mean results indicate 12 out of the 15 largest outlet glaciers reach maximum LGM thickness and extent synchronously, at around 25.1 ka (see Supplementary Fig. 10), when our climate forcing produces peak cooling (Fig. 2e). Four main outlet glaciers however display an asynchronous response (see Supplementary Fig. 10). These include the Dora-Baltea and Dora-Riparia outlet glaciers, which reach maximum extents eight and three centuries before other sampled outlet glaciers, respectively. The modelled Rhône outlet glacier reaches maximum extent around 24.6 ka, five centuries after most glaciers (including the nearby Rhein outlet glacier) started to retreat from their maximum positions. In all NROYS, the Lyon outlet glacier remains near its maximum extent until 24.6 ka, in line with recent geochronological evidence[6] (see Supplementary Fig. 11). Results thus suggest the Lyon and Rhône outlet glaciers were impacted by notable mechanisms of inertia during the LGM,

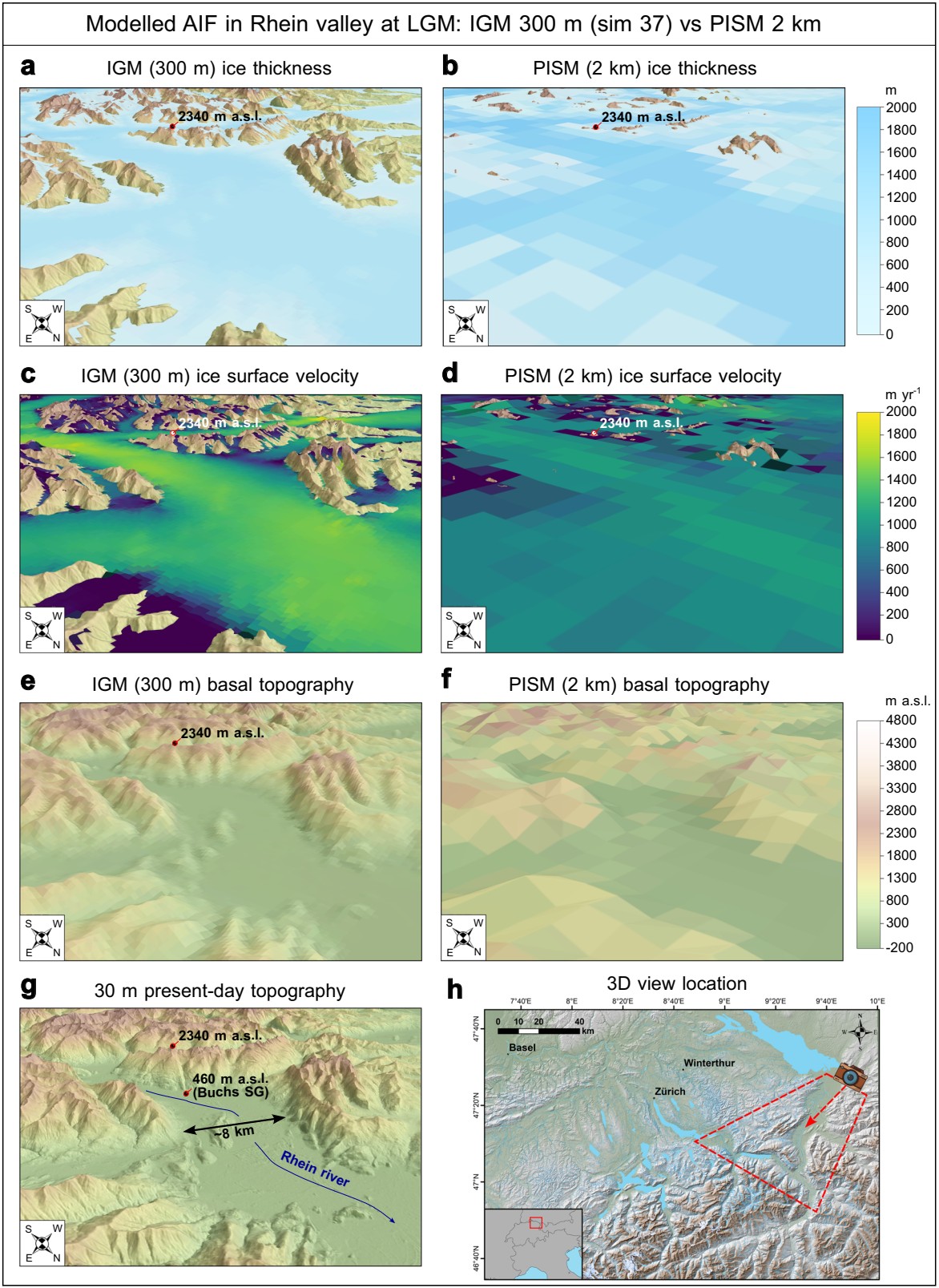

**Fig. 5 | Three-dimensional views of best-fit Instructed Glacier Model (IGM) 300 m simulation (number 37).** Here, the simulation is compared against the Parallel Ice Sheet Model (PISM) 2 km simulation by Jouvet et al.[22], and displayed by showing modelled Last Glacial Maximum (LGM) ice thickness (**a**, **b**), ice surface velocity (**c**, **d**), and basal topography (**e**, **f**) fields in the main Rhein valley (Alpenrhein). Ice thickness and velocity fields are plotted above a 30 m digital elevation model of the local topography, also shown in **g**. **h** Indicates the location and direction of the three-dimensional view shown in other panels. More versions of this figure are available for other regions of the Alps in Supplementary information. All panels were produced using ArcGIS Pro 3.2.2 (Esri), and panels **a**–**d**, and **g**, **h** include the 30 m digital elevation model from ALOS World 3D DEM[82] (version 4.1). **h** Features geographical data from OpenStreetMap.

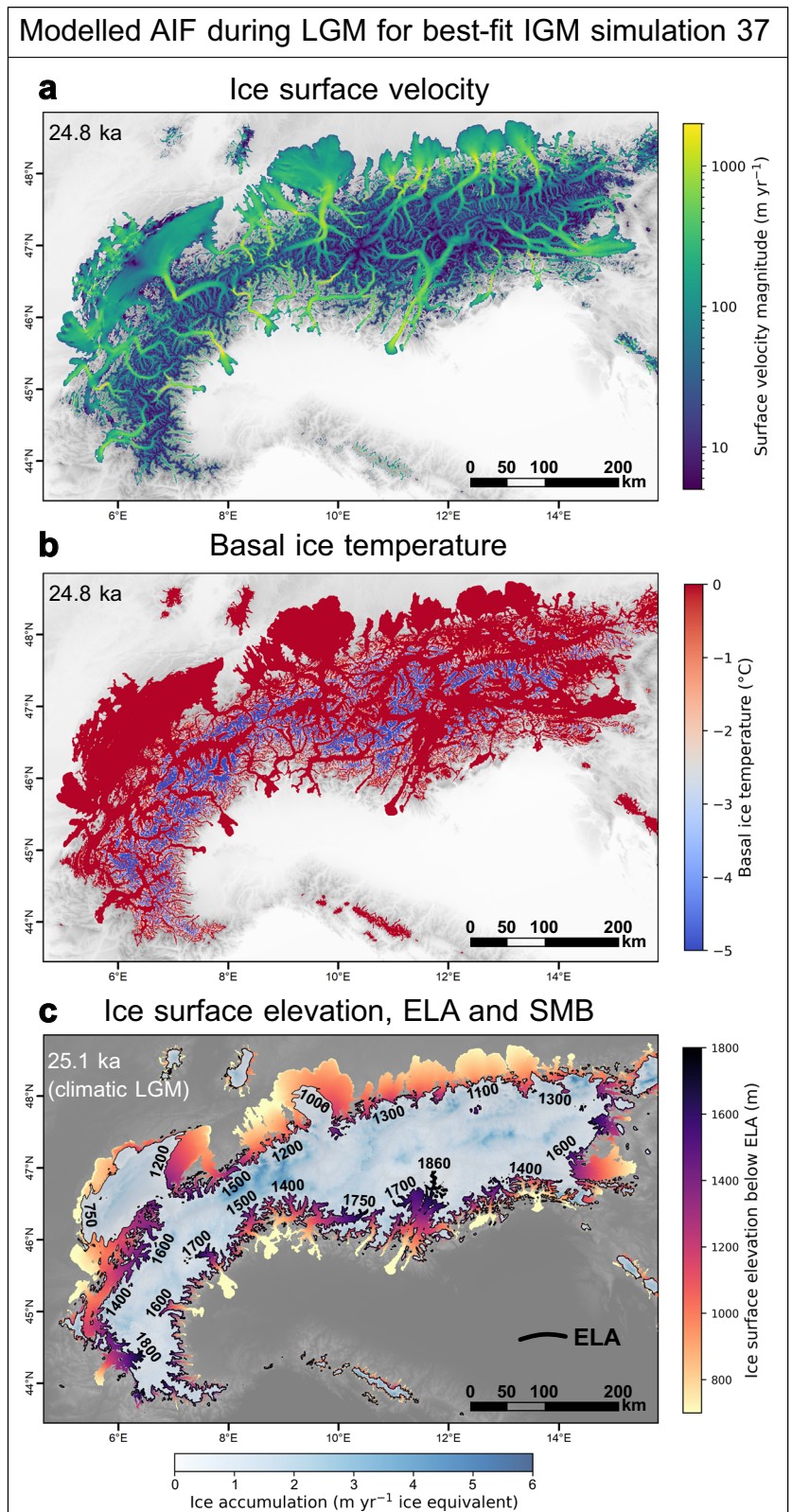

**Fig. 6 | Alps-wide ice velocity, basal temperature, and Equilibrium Line Altitude (ELA) from our Instructed Glacier Model (IGM) results.** Are displayed: the modelled Last Glacial Maximum (LGM: 24.8 ka) fields of ice surface velocity (panel a), pressure-adjusted basal ice temperature (**b**), and ELA during the climatic LGM (25.1 ka) along with ice surface elevations of the Alpine Ice Field ablation zone, and Surface Mass Balance (SMB) in accumulation zone (**c**). In **c**, black numbers indicate the average ELA (in m a.s.l.) for the highlighted sectors.

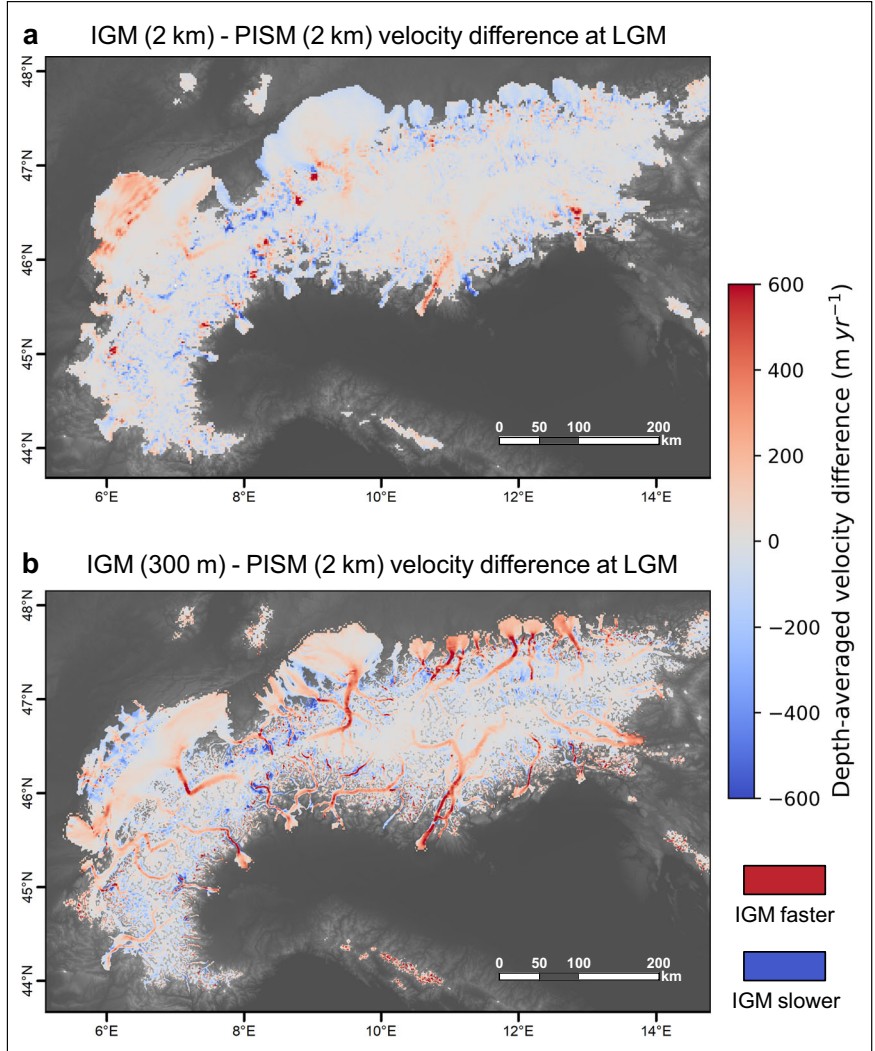

**Fig. 7 | Alps-wide differences in modelled ice flux between 300 m and 2 km.** Panel a shows the differences in depth-averaged ice velocity (ice flux) at the Last Glacial Maximum (LGM) between the Instructed Glacier Model (IGM) ran at 2 km resolution (see 'Methods' section: 'Model validation') and the Parallel Ice Sheet Model (PISM) 2 km simulation of Jouvet et al.[22]. To visualize the impact of increasing model spatial resolution on ice velocities and ice flow patterns; panel b shows the same velocity difference but with IGM ran at 300 m resolution instead (ensemble best-fit simulation 37).

likely related to their longer ice flow paths causing slower enthalpy-induced responses of margin extents to changes in air temperature. Such modelled asynchronies are likely too short-lived to be detected by dating techniques (e.g. surface exposure dating) which, at the LGM, typically produce analytical uncertainties of 0.8–2.0 kyr. We note previous AIF-wide models[20,22] produced greater inter-glacier asynchronies (up to 3-4 kyr) than our NROYs. This may be due to these previous studies[20,22] using a different glacial index signal (from Antarctic ice core[40]) but also to coarser-resolution models producing thicker AIFs, with lower surface slopes and lower velocities in main draining valleys, enabling possible delays in flux adjustment to climate variations. Moreover, a thicker modelled ice field can generate greater and more spatially-heterogeneous climate-response times due to a slower enthalpy feedback on basal ice temperatures and sliding.

The improved model-data fit also enables us to better constrain the former AIF surface mass balance field during the LGM, with implications for discussing the LGM climate of the European Alps. Here, all our NROYs consistently produce surface mass balance fields with high inter-regional variability in equilibrium line altitudes during the climatic LGM (~25.1 ka with our data), with a maximum difference of up to ~1.1 km between the southwestern Jura (~750 m) and upper

Garda (~1860 m) catchments (Fig. 6c). NROY results suggest north-to-south increases in equilibrium line altitudes (of up to 850 m) in nearly all regions of the former ice field. One exception is the Aare and Western Ticino-Toce catchments, which display similar equilibrium line altitudes (~1500 m) (Fig. 6c). On average, we find that during the LGM, modelled equilibrium line altitude gradients between northern and southern glaciated regions of the Alps are either similar or greater than regionally equivalent gradients reported for the present-day (~2000 AD)[64]. Our modelled surface mass balance fields are thus incompatible with the recurring hypothesis of a shift from the contemporary westerly-dominated to a more southerly-dominated moisture advection pattern during the LGM[43], which would lead to lower north-to-south equilibrium line altitude gradients.

The present-day uplift signal in the European Alps is also subject to a long-lasting debate[65,66], as it results from a combination of mechanisms acting over various timescales ($10^3$ to $10^7$ years), and whose relative contributions remain uncertain. These drivers include erosional unloading, tectonic deformation, lithospheric slab processes, and postglacial rebound[21,67]. Process-based uncertainties further arise from a lack of crustal shortening detection in the Western

and Central Alps[68]. Our AIF model has implications for better constraining the contribution of one of the drivers, i.e. postglacial rebound. A thinner ice field (Fig. 3) necessarily generates less glacial isostatic adjustment. During the LGM, we find our NROYs produce a maximum crustal deflection near the AIF centre of $93.1 \pm 13.5$ m, a value between 1.5 and 3 times lower than previous glacier-modelling-derived estimates (150–280 m[21,22]), despite using variable lithospheric effective elastic thickness values (see Supplementary Figs. 5, 12, 'Methods' section). This may imply a smaller postglacial rebound than previously proposed, although we acknowledge uncertainties in the upper mantle architecture (lithospheric mantle and asthenosphere) preclude any definitive answers. Moreover, the viscous, time-dependent component of the deformation, which causes a delay between crustal (un)loading and the flexural response, is not considered here as our isostatic model (gFlex) only accounts for the elastic deformation obtained after all crustal loads have been entirely compensated[39]. Our model could however inform future investigations using more complex earth deformation models to better assess the contribution of former ice unloading to the uplift signal observed today in the European Alps.

The approach presented in this study also has implications for improving the reconstruction of other Quaternary ice fields and ice sheet histories. Indeed, achieving spatial resolutions required for a data-consistent LGM model of the AIF was previously unattainable with traditional central-processing-unit-based models. This study demonstrates that physics-driven machine learning and GPU-based processing makes simulating a continental-scale ice field at high (300 m) resolution with a thermo-mechanically coupled, three-dimensional, and high-order model, possible. This is moreover achieved at a fraction of traditional models' computational costs and with identical parameter modularity. We thus argue that physics-informed GPU-based models open the door to a new era of more accurate paleo ice field and ice sheet modelling. In particular, the ability to more accurately model the complex dynamics of topographically-constrained large-scale ice fields, such as the AIF during the LGM, through high-order modelling, higher spatial resolutions and wider ensemble-type explorations of parameter spaces, will likely reduce model-data misfits in many other studied mountain ranges. Investigations that would likely benefit from this modelling approach include the reconstructions of former large-scale ice fields during Quaternary glaciations in Patagonia, Alaska, high-mountain Asia, Turkey, Georgia, California, Peru, the Pyrenees, New-Zealand, and Iceland, to name only a few. This improvement in modelling capabilities should also spark new motivation to collect more field data to better constrain the former evolution of ice extent (e.g. with more detailed terminal and lateral moraine mapping and dating[5]), ice thickness and thinning rates (e.g. with new trimlines and cosmogenic nuclide dipstick data[69]), and ice internal flow direction and dynamics (e.g. with lineation mapping and flow set reconstructions[70], new erratic transport and provenance databases, borehole data and englacial stratigraphic reconstructions[71,72]) of other former glaciers, ice fields and ice-sheets. This has implications for not only scientific discoveries in paleoglaciology, but also across other disciplines studying processes linked to Quaternary glacial history, including archaeology, paleoclimatology, paleoecology, and geomorphology.

We also believe this technology may permit more accurate simulations of past and future Greenland- and Antarctic-ice-sheet change, by facilitating ice-sheet-wide simulations at considerably higher resolutions than previously achieved and at lower computational costs. This would allow for broader explorations of parameter spaces and ice-sheet evolution scenarios, with implications for more accurately modelling basal conditions and ice-ocean interactions towards grounding lines, crucial for better projecting the future response of continental ice sheets to climate change[38]. Moreover, reducing Quaternary model-data misfits, as is achieved here with the AIF, may be important for improving the paleo initialisation procedures of Antarctic and Greenland ice sheet models, which greatly impact their future projections[38,41]. Therefore, we expect physics-informed, GPU-based models to bring advances in projecting future global sea level rise and environmental change associated with the response of contemporary ice sheets to climate change.

To conclude, we believe the development and application of physics-driven AI and GPU-based glacier and ice-sheet models should become a primary avenue of future research due to its wide and multidisciplinary implications for the fields of Quaternary and cryosphere science.

## Methods

### The glacier model

The evolution of glacier ice thickness, denoted as $h(x, y, t)$, starting from an initial glacier shape, is governed by mass conservation, which connects elevation change, ice dynamics, and mass balance through the following continuity equation:

$$\frac{\partial h}{\partial t} + \nabla \cdot (\bar{\mathbf{u}} h) = \text{MB}, \tag{1}$$

where $\nabla$ represents the divergence operator for the horizontal variables $(x, y)$, $\bar{\mathbf{u}} = (\bar{u}, \bar{v})$ denotes the vertically-averaged horizontal ice velocity field, and MB represents the surface and basal mass balance functions. Equation (1) is solved using an explicit upwind finite volume scheme on a regular grid which allows the model to update the ice thickness while conserving mass. In the following sections, we describe in turn individual IGM[31,32] sub-models used in this study for simulating processes of ice flow, ice enthalpy, climate forcing, surface mass balance, isostatic adjustment, and avalanching.

### Ice flow

Ice flow dynamics is modelled using Glen's flow law:

$$D = A\tau^n \tag{2}$$

where $D$ and $\tau$ denote the strain rate and deviatoric stress tensors, with $A$ representing the rate factor, and $n = 3$ as Glen's exponent. Here, we use the Blatter-Pattyn model[33], which disregards second-order terms in the thickness/length ratio in the momentum conservation equation. This modification makes solving the stress balance easier than with the original Full-Stokes model. For our boundary condition, we use a nonlinear Weertman friction condition (e.g. Schoof & Hewitt[73]), relating the basal shear stress $\tau_b$ to the sliding velocity $\mathbf{u}_b$ as follows:

$$\tau_b = -c|\mathbf{u}_b|^{m-1}\mathbf{u}_b, \tag{3}$$

where $m > 0$, $c = c(x, y) > 0$ is the sliding coefficient.

Rather than using a traditional solver, IGM models the ice flow using a physics-informed deep learning emulator[74] trained to minimise the energy associated with the Blatter-Pattyn equation[32]. Thanks to its efficient evaluation and training on GPUs, the neural network considerably reduces computational cost with negligible accuracy losses relative to a traditional ice-flow solver[32]. The neural network features 16 two-dimensional convolutional layers representing 140, 000 trainable parameters. We use the hyper-parameters found by Jouvet et al.[31]. To obtain initial weights and facilitate convergence, the neural network is pretrained over a diverse catalogue of glaciers and flow regimes[32]. Moreover, it is frequently re-trained during transient IGM simulations (every seven iterations) to adjust to new glacier states obtained through time. This frequency was found to be a good trade-off maintaining accuracy while keeping computational cost relatively low.

## Ice enthalpy

Temperature within the ice is modelled with an energy-conservative enthalpy model following Aschwanden et al.[34]. Ice enthalpy, $E$, is a function of ice temperature, $T$, and ice water content, $\omega$:

$$E(T, \omega, p) = \begin{cases} c_i(T - T_{ref}), & \text{if } T < T_{pmp}, \\ E_{pmp} + L\omega, & \text{if } T = T_{pmp}, \ 0 \leq \omega \end{cases} \quad (4)$$

where the temperature, $T_{pmp}$, and enthalpy, $E_{pmp}$, at pressure-melting point of ice are defined by

$$T_{pmp} = T_0 - \beta p, \quad (5)$$

$$E_{pmp} = c_i\left(T_{pmp}(p) - T_{ref}\right) \quad (6)$$

According to the definition of enthalpy prescribed above, we have two possible modes: i) When the ice is cold (i.e. below the melting point), the enthalpy is simply proportional to the temperature minus a reference temperature. ii) When the ice is temperate, the enthalpy continues to increase. In this case, the additional component, $L\omega$, accounts for the creation of water content through energy transfer. The enthalpy model consists of an advection-diffusion equation, with horizontal diffusion being neglected, and strain heating and drainage as source terms. At the modelled ice surface, the enthalpy equation is constrained by the surface temperature provided by the climate forcing. Borehole data show the offset between temperatures of surface air and of the active ice layer at glacier surfaces is challenging to constrain and varies through both space and time[75]. Here, we set this offset as an ensemble-varying parameter with possible values ranging between 1 °C and 3 °C (surface ice being slightly warmer), bracketing the median (1.85) of observed offset values ($n = 41$) reported by Zagorodnov et al.[75]. At the glacier bed, there are multiple boundary conditions for the enthalpy equation depending on the bottom ice layer and bed surface temperature[34], the latter being forced, in this study, by geothermal heat flux data from Goutorbe et al.[35]. The 3D advection-diffusion equation is solved using a semi-implicit scheme with finite differences at each time step, defined by the time stepping of the mass conservation. As a result, the ice enthalpy, temperature, and basal melt rate are all updated at each time step. The enthalpy impacts both internal ice flow by influencing the rate factor, $A$, via the Glen-Paterson-Budd-Lliboutry-Duval law[56]:

$$A(T, \omega) = A \exp(-Q/(RT_{pa}))(1 + 181.25\omega), \quad (7)$$

and basal sliding, $c$, via meltwater production when pressure melting point is reached. Our implementation of the enthalpy formulation in IGM successfully passed the two benchmark experiments proposed by Kleiner et al.[76] and Hewitt & Schoof[77]. These tests, as well as more details regarding the enthalpy model can be found alongside IGM's source code (https://github.com/jouvetg/igm).

## Sliding parameterization

Following Bueler & van Pelt[78], the basal water thickness in the till layer, $W_{till}$, is computed from the basal melt rate, $m_b$, obtained from the enthalpy as follows:

$$\frac{\partial W_{till}}{\partial z} = \frac{m_b}{\rho_w} - C_{dr}, \quad (8)$$

where $C_{dr}$ is a simple drainage parameter. The till layer is assumed to be saturated when the basal water layer thickness reaches a caping value of $W_{till}^{max} = 2m$. The effective thickness of water within the till layer $N_{till}$ is computed from the saturation ratio $s = W_{till}/W_{till}^{max}$ by the

formula[78]:

$$N_{till} = \min\left\{ p, N_0 \left(\frac{\delta P}{N_0}\right)^s 10^{(e_0/C_c)(1-s)} \right\}, \quad (9)$$

Where $p$ is the ice overburden pressure and the remaining parameters are constant. The sliding coefficient, $c$, in (3) is defined by the Mohr-Coulomb law[56] that involves the effective pressure in the till $N_{till}$:

$$c = \tau_c u_{th}^{-m} = N_{till} \tan(\phi) u_{th}^{-m}, \quad (10)$$

where $\phi$ is the till friction angle[25]. Here, using the assumption that basal materials are generally weaker (softer sediments) in valley troughs than over mountain tops[38], $\phi$ is parameterised to be a piece-wise linear function of bed elevation, $b$:

$$\phi(x,y) = \begin{cases} \phi_{\min}, & b(x,y) \leq b_{\min}, \\ \phi_{\min} + (b(x,y) - b_{\min})M, & b_{\min} < b(x,y) < b_{\max}, \\ \phi_{\max}, & b_{\max} \leq b(x,y). \end{cases} \quad (11)$$

Between ensemble simulations, we modify the elevation-dependency of $\phi$ by varying upper and lower bed elevation thresholds ($b_{\min}$, $b_{\max}$) between ranges of (−500, −100) metres, and (2400, 3000) metres, respectively, while till friction angle thresholds ($\phi_{\min}$, $\phi_{\max}$) are set to values of 15 and 50 (see Supplementary Table 2). To further modify the sliding coefficient for a given basal shear stress, the parameter $u_{th}$ also varies within our ensemble between 100 and 2000 m yr⁻¹.

## Climate forcing

In this study, input climate forcing fields are taken from Jouvet et al.[22]. First, the Weather Research and Forecasting regional climate model[27] was used to downscale time slice simulations of a global Earth system model[79] to high-resolution (2 km) over the European Alps. This workflow produced climate snapshots, including weekly mean and standard deviation data, for the pre-industrial (1850 AD), LGM in the Alps (~24 ka) and Marine Isotope Stage 4 (65 ka) periods (see Figs. 5–7 in Jouvet et al.[22] for more details). Here, we extend climate data between these three snapshot states using a glacial index approach (e.g. Niu et al.[41]). This method creates continuous climate fields with two given states: the pre-industrial climate snapshot with limited ice cover in the Alps, and the LGM climate snapshot. The glacial index function is built by linearly rescaling a climate proxy signal such that the glacial index is close to 1 at the LGM and close to 0 at the pre-industrial. In this study, we use an Alps-specific climate proxy signal as input to our glacial index scheme (Fig. 2e), which combines the Bergsee lacustrine record[42] (35–30 ka: Black Forest, Southern Germany) and the Sieben Hengste speleothem $\delta^{18}O$ record[43] (30–18 ka: Bernese Alps, Switzerland). Note that input air temperature fields refer to the surface topography given as input to the regional climate model, which features glaciers at their maximum extent for Marine Isotope Stage 4 and the LGM states, and the present-day topography for the pre-industrial state. To simulate the temperature when the modelled surface deviates from the reference one, we apply a vertical and linear correction using an atmospheric lapse rate of 6 °C km⁻¹.

While the above climate forcing improved the overall model-data fit in the LGM extent of the AIF relative to previous studies[20,24], certain outlet glacier extents remained either too large or too small despite varying non-climatic parameter extensively. We assume these isolated misfits are likely related to uncertainties in the input climate and/or limitations of the glacial index approach, and thus implement an ensemble-varying and glacier-catchment-specific precipitation offset scheme. To do so, we apply time-independent scalar multipliers to the input precipitation field over the Rhein (reference multiplier = 0.9), Isar (1.33), Jura (0.7), Drau (1.33) and southernmost

Alps (0.7) regions. In other regions, precipitation remains equal to the original input field. The magnitude of these offsets is made simulation-dependent by using an ensemble-varying multiplier parameter (ranging between 0.85 and 1.15) that further modifies reference offset values. Original precipitation values can thus be modified by between -40% and +50%, depending on the region and ensemble simulation (See Supplementary Table 1).

## Surface mass balance

In this study, we parameterize IGM with a combined snow accumulation and positive degree-day model[80] to compute surface mass balance from input temperature and precipitation fields. In this scheme, precipitation generates surface accumulation (falls as snow) when air temperature is below 0 °C and causes no accumulation (falls as rain) when temperature is above 2 °C, with a linear transition in between. Surface ablation, on the other hand, is computed proportionally to the number of positive degree days. The positive degree day integral is numerically approximated using week-long sub-intervals, following Calov & Greve[36]. Positive degree day parameters are not well constrained and can vary in space and time. Thus, in our ensemble, the melt factor for ice is simulation-dependent and varies between 6 and 9 mm w.e. $d^{-1}$ °$C^{-1}$ (see Supplementary Table 1). The melt factor for snow remains constant at 3 mm w.e. $d^{-1}$ °$C^{-1}$. Our positive degree day scheme also models the refreezing (turned into net accumulation) of a given proportion of the computed melt. This proportion is here made simulation-dependent and varies between 50% and 70% within the ensemble (see Supplementary Table 1). Note that no proglacial lake module is implemented in this model setup, meaning that all ice is assumed to be land-terminating. We consider this assumption to have little impact on the LGM geometry of the AIF since most overdeepened basins of the Alpine foreland were eventually ice-filled during maximum glacier advance[18].

## Model initialisation

All IGM ensemble simulations are initialised with ice-free conditions at 35 ka. Although unrealistic, a sensitivity analysis reveals that starting an ice-free, AIF-wide simulation at 40 ka instead does not impact modelling results at the LGM, as diagnostic model variables converge after 4–5 kyr of running the model (see Supplementary Fig. 13). Therefore, as suggested by previous modelling work[22], the response time of the AIF to climate change during the last glaciation does not exceed 4–5 millennia.

## Model validation

Before applying IGM and conducting AIF-wide simulations at 300 m resolution, we quantitatively assessed the model's suitability and fidelity in simulating the last glaciation of the European Alps. To do so, we ran simulations at the same 2 km spatial resolution as Jouvet et al.[22] to quantitatively compare outputs with PISM[25], a widely used and well-tested model[25]. Unlike our higher-resolution (300 m) ensemble runs (see 'Results' section), these test simulations use the same glacial index climate signal (EPICA ice core[40]), basal topography, and space-independent basal till friction angle ($\phi$ = 30°) parameterization as Jouvet et al.[22]. Furthermore, no avalanche module is employed in this 2 km setup. This experiment resulted in a 2 km simulation with IGM producing AIF extents, volumes, ice velocities, surface mass balances and basal conditions that are consistent with PISM outputs from Jouvet et al.[22] (see Supplementary Figs. 14–16). Between the two models, maximum AIF volume and areal extent (reached in this case at ~24.5 ka) vary by less than 2%, while LGM ice thickness differences remain minimal (−6 ± 147 m). At 2 km, Alps-wide differences in LGM ice flux between IGM and PISM are minor with surface, basal, and depth-averaged velocities varying by 2 ± 110 m $yr^{-1}$, -27 ± 110 m $yr^{-1}$, and −6 ± 108 m $yr^{-1}$, respectively. This test shows that IGM can be used to model the AIF's last glaciation

with results comparable to a well-established ice-sheet model (see Supplementary Figs. 14–16). Results are not expected to be 100% identical, however, as important differences remain between the two models, the most potent of which is the ice flow stress balance, approximated with the SSA/SIA hybrid model in PISM[25], and with the higher-order Blatter-Pattyn[33] model in IGM[32].

## Isostatic adjustment

In the European Alps, the lithospheric deflection caused by ice loading during the LGM was large enough to notably influence ice surface slope and elevation, and thus ice flow dynamics and mass balance (see Supplementary Fig. 12). To account for space- and time-dependent bed deflection, we couple IGM with the gFlex model[39] which dynamically computes the flexural isostatic adjustment using the two-dimensional elastic thin-plate Eq. (1) in Mey et al.[21]. For gFlex-specific calculations, the frequency of iteration is here set to 50 years while the spatial resolution remains at 2 km. At the entire Alps scale, the impact of a higher frequency or spatial resolution on modelled AIF evolution is negligible[81]. For our 300 m resolution IGM runs, we feed gFlex a space- and time-independent lithospheric effective elastic thickness, representing the resistance to bending under specific vertical loads. This elastic thickness is challenging to constrain and is thus set as an ensemble-varying parameter ranging from 35 to 50 km, after Mey et al.[21] (see Supplementary Table 1).

## Avalanche scheme

Increasing spatial resolution with IGM produces finer and steeper bed topographies in upper alpine catchments and accumulation zones. Accurate modelling of transient glacier evolution on such topographies requires an approximate representation of avalanching which impacts ice accumulation, surface elevations, and flow velocities. With IGM, we thus make use of an avalanche module that redistributes modelled accumulation downslope until the glacier surface reaches a given angle of repose value, here set to 45° across the domain and for all simulations. A sensitivity analysis on the best-fit simulation (number 37) reveals that using a value of 35°, on the low-end of reported angle of repose values for Alpine glaciers, does not generate a notable impact on AIF-wide model-data fit in LGM ice extent (−0.03%) and thickness (−3%), relative to 45° (see Supplementary Fig. 17). In our simulations, the frequency of the avalanche module updates is set to 5 years.

## Empirical trimline elevation data

Field-based estimation of trimline location and elevation is non-trivial, yields geomorphological uncertainties, and can sometimes be challenged by dangerous access conditions forcing remote observations. Therefore, in this study, empirical trimline elevation data ($n$ = 396), gathered from literature[10–16], are quality-controlled by comparing them against independent elevations extracted from high-resolution (≤ 5 m) digital elevation models at their reported locations (see Supplementary Table 3). Given the relatively low (<10 m) vertical error of these satellite-derived products, a trimline data point yielding a >50 m offset between the two independent elevations was here considered an outlier likely due to error in measurement of either trimline geolocation or elevation. Trimlines yielding >50 m offsets were found to represent ~11% of the original dataset ($n$ = 43 out of 396). The impact of removing these 43 presumed outliers on the mean model-data misfit between modelled ice surface elevations and reported trimline elevations is negligible (on average a ~2% reduction). However, it notably reduces the standard deviation of this misfit, on average by ~40%, thus decreasing scatter and generating a notable improvement in model-data ice thickness agreement. Details of the different digital elevation models used for this analysis are listed in Supplementary Table 3.

## The empirical LGM outline of the AIF

The empirical LGM outline of the AIF used in this study was originally produced by an Alps-wide compilation of geomorphological and geochronological evidence by Ehlers et al.[17]. Since this key study, the outline has been updated by a series of empirical investigations improving the quality of LGM margin reconstructions in specific sectors of the Alpine foreland. In this work, we use the most up-to-date version of the outline. Here, we provide a summary of the main studies which have updated the LGM AIF outline since Ehlers et al.[17]. Gianotti et al.[50,51] have made updates to LGM margins of the Ivrea outlet glacier in the Dora Baltea region. Braakhekke et al.[49] have done so for the Orta region, Kamleitner et al.[5] for the Verbano region, Ravazzi et al.[53] for the Oglio region, Monegato et al.[48] for the Garda region, Federici et al.[47] for the Gesso region, Ribolini et al.[8] for the Stura region, and Ivy-Ochs et al.[52] for the Dora Riparia region.

## Data availability

The data that support the findings of this study are available in the supplementary information document, and from the corresponding author upon request. Additionally, the data presented in the paper figures, along with a catalogue of videos displaying Alps-wide and regional results from our best-fit model simulation, are available from the following open-access online repository: https://doi.org/10.5281/zenodo.14275231.

## Code availability

The Instructed Glacier Model (IGM) source code (Python programming language) and documentation are available from Guillaume Jouvet's GitHub repository at https://github.com/jouvetg/igm. Codes, IGM version, and instructions required to reproduce this specific study's experiment and results are available from the following open-access online repository: https://doi.org/10.5281/zenodo.14275231.

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

## Acknowledgements

This research was funded by the Swiss National Science Foundation, through a grant (RECONCILE: project number: 213077) awarded to G.J, A.V., and S.N. J.M. acknowledges funding by the German Academic Exchange Service under the Project-ID 57553291. This work benefited from discussions with Prof. Christopher Clark, Dr. Jeremy Ely, Dr. Remy Veness, Dr. Julien Seguinot, Dr. Rose Archer, Dr. Samuel Cook, Kejdi Lleshi, Dr. Maxime Bernard, Prof. Georgina King, and Dr. Jordi Bolibar. We thank Prof. Christoph Raible, Dr. Emmanuel Russo, Dr. Jonathan Buzan for producing the paleoclimate snapshot simulations. We thank Dr. Jerzy Zasadni for providing trimline observation data.

## Author contributions

T.P.M.L. and G.J. conceived and designed the study with the help of S.K. and A.V. G.J. developed and improved IGM to build a model setup ready for this specific study, with feedback from tests by T.P.M.L. T.P.M.L. carried out the modelling work, model-data comparison, wrote the paper, and produced the figures together with G.J., S.K., and F.H. S.K. compiled and quality-controlled trimline observation data. J.M. provided the valley-fill-sediment-removed topography and developed the isostasy and avalanche IGM modules. T.P.M.L., G.J., S.K., A.V., F.H., S.I.O., J.M., B.D.F., A.H. and S.N contributed to the discussions, interpretations and writing of the manuscript.

## Competing interests

The authors declare no competing interests.
