## [Transparent Peer Review file · Nature Communications]

A data-consistent model of the last glaciation in the Alps achieved with physics-driven AI

Corresponding Author: Dr Tancrède Leger

Version 0:

Reviewer comments:

Reviewer #1

(Remarks to the Author)

Review of Leger et al: a data consistent model of the last glaciation in the Alps achieved with physics-driven AI

Thank you for the opportunity to review this paper. This manuscript seeks to improve the modelling the European Icefield using the Instructed Glacier Model. The use of GPUs means that processing is fast and efficient, allowing higher spatial resolution across the domain. The ability to model icefield-wide at 300m resolution is an extraordinary achievement. This is high enough to capture the deep, narrow alpine valleys, a major control on ice flow and ice dynamics that are typically neglected by more computer intensive models, and the use of higher order Blatter-Pattyn physics ensures that the most important stresses are accounted for. The model-data comparison framework to score the ensemble is state of the art. The results of the paper and especially the methodological steps forward gained have far-reaching implications.

The surface mass balance scheme used is pretty simple however and perhaps this is the next area that the authors could tackle to improve!

This manuscript is an exciting paper that I enjoyed reading. It is truly transformative science at its best and a major leap forwards. My congratulations to the authors. I have only a few suggestions, mostly for clarity, and to help the paper fulfil its potential as a highly cited transformational piece of work.

Mostly these comments are to do with: design of the model-data comparison; and the wider implications in the discussion.

Model data comparison:

The empirical outline used in figure 1 is used somewhat uncritically and I think more discussion is needed (perhaps in the Methods or Supplement) more analysis of this limit. What is the uncertainty in the limit? What ages are the moraines and are they all the same age? When you score your model based on the best fit to moraines, this assumes that all moraines are the same age and are well dated. However, we know that topography has a significant influence on moraine formation. Where you have model underfit (e.g. GA glacier), what is the confidence in the age of the moraines?

The Ehlers limit is fairly dated now and on reading it wasn't clear to me if this was used as published or if it was updated based on the more recent works cited. This needs to be clearer as although trimlines are used largely to score the ensemble the moraines are used to find the best fit scenario. How well dated are the moraines and the trimlines?

Trimlines are also challenging to date – often boulders and erratics here will be transported supraglacially and therefore have an inheritance when used in cosmogenic exposure age dating. In the UK for example, trimlines have proven hard to date and age is debated.

Overall, some more discussion somewhere about the data used, the ages of the moraines, and how much confidence there is in it would be helpful. This could be in the Methods and SI. I also wonder if this may help explain where you have under/overfit.

The 'quality controlled' reference to trimlines in caption of figure1 – this was not clear to me what this entailed.

Wider implications:

I wondered about the wider implications of the work and think that the authors could do more agenda setting in the discussion. The Discussion is rather inward looking and the wider implications to a wider audience can come across more clearly.

The use of trimlines to realistically constrain the model ensemble members is exciting and I can see this spawning a new generation of mountaineer glaciologists climbing peaks to obtain cosmogenic nuclide ages. If the use of trimlines rather than moraines (more commonly used) to evaluate the ensemble makes for a systematically better product, then you can highlight this to glacial geologists and indicate the kinds of data needed for model-data comparisons. I think that this would bring the

work to the forefront of a wider audience and would increase readership and applicability beyond the modelling community. Emphasise the need for more well-dated, well constrained timeline data in those parts of the world with currently rapidly changing icefields (Canada, Alaska, Patagonia, Himalaya, Antarctic peninsula), where the impacts of glacier recession will be significant, and highlight that constraining past rates of thinning would be especially beneficial to improving future projections. The kinds of data needed could be more clearly signposted.

The perturbed ensemble approach to maximise model-data fit will be critical in forward modelling these icefields and demonstrates the importance of hindcasts and data model integration to calibrate projections. This can be emphasised more – there is a rather vague reference to Greenland and Antarctica in the conclusion but here, the thicker ice sheets have fewer nunataks and it is likely the smaller mountain icefields more similar to the Alps that this approach will be most beneficial. Is the model set up useful for forward modelling of the Alps?

Minor comments

I wondered how lakes that formed in over deepenings were accounted for and if frontal ablation was simply neglected. Does the model assume that all glaciers are land terminating? Frontal ablation is likely to be a significant driver of ice dynamics so although this is not an issue to be corrected it would warrant a minor sentence or two to clarify.

In the experiment design (line 218) please clarify: are the parameter spaces weighted or selected according to probability or known ranges?

Line 4 -7: It is helpful to cite some references for mentioned 'an abundant library of mapped and dated ice-contact glacioenic landforms and sediments separately for reader's interest.

Line 15-17: Name the glacier evolution models (for example with some references)

Line 39: What is the transient time-step of model simulation results, is it the same as the temporal resolution of climate dataset?

Line 169: (I would expand more on ensemble for clarification: is it ensemble of 100 parameterized model simulations resulting from a group of parameter sets and /or a group of climate input used in the study)

how is the initial range for each parameter selected? Is it from the output of the model simulations or based on the model default values or from literatures?

Line 225 -227: Any explanation for why the maximum AIF areas outside the range of 156-165X 103 produce substantial under-or overestimations of the LGM ice extent reconstruction?

Line 228- 230: Could it be because of the deep learning algorithm that the model is producing similar patterns for outlet glaciers despite extensive changes in the parameters? does this also mean these outlet glaciers control the evolution and behaviour of the entire AIF?

Line 241-243: how is this natural break for mean and standard deviation i.e. 180 m and 100 m in elevation difference chosen? (is it from sieved simulation data set?)

Line 247: the best fit to moraines ensemble member selection: was how was this quantified? Visually?

Line 280 – unprecedented is a very strong word, can it be justified? I would avoid using more superfluous language personally.

Line 326-327: why is that so? Is it because this outlet glacier is influenced by more local phenomena than the Alps-wide phenomenon and the ensemble parameter sets could not capture the processes and drivers?

Line 408: explain briefly what high-magnitude venturi effects mean (not necessarily all glaciologists know what it means)

Line 557 – how good is the chronological evidence here?

Line 637 – how else could it inform future investigations – see comments above.

Line 647 – there are also implications for data as well

Line 656 – this is very vague and I suggest rewriting this paragraph. Greenland & WAIS/EAIS have very few nunataks and requires a different approach. Can a very small number of spatially limited dispersed nunataks improve modelling here?

Line 659 – a rather superficial sentence to end on. You can finish with a more exciting call to arms and set the agenda for the community here.

Some limitations and how they can be improvised have been explained in the discussion sections (Line 630- 637).

However, it would be beneficial to clarify the major limitations of such high-resolution, 3D -AI-based models and outline the key considerations when integrating multiple physical processes into a single, high-functioning model.

Discussion

This section is well-explained and effectively discussed in relation to previous research. It offers valuable new insights into both glacial dynamics and landscape evolution. The revelation of localized ice flow speedups in narrow valleys and topographic bottlenecks, along with rapid valley deepening in areas like the Rhône valley, highlights the critical role of high-resolution, physics-informed models in addressing unresolved questions in large-scale past glacier modelling. Additionally, the inclusion of future research perspectives and avenues with the IGM model is appreciated. This will be more useful for the cryosphere community and quaternary researchers.

Methods

More information on model ensemble evaluation would be useful here.

More information on the data used for model evaluation would also be useful.

Can IGM be advised to use for reconstructing a single glacier elsewhere in the world with same ensembled parameter sets?)

(Remarks on code availability)

Reviewer #2

(Remarks to the Author)

Review of Leger et al

This study presents a major breakthrough in glacier modeling capability. New information and understanding gleaned from paleo-glacier simulations, and future glacier simulations, is most robust when models run in ensemble mode, that is, large numbers of simulations that allow one to account for a range in model inputs. Ensemble results also provide a way to characterize the uncertainty of numerical simulations. But running ensembles does not occur frequently because most numerical glacier models are too computationally demanding. This study relies on emulators and other machine learning techniques to speed up the modeling process, and allows for the ensemble approach.

The authors test this model capability in an oft-used target for glacier modelling (the European Alps) and builds on recent work that provides context for the new cutting edge approach. This nicely described and illustrated study has me convinced that it is deserving of publication with minor revisions.

I have some comments more significant than others, these I keep in the line-by-line section below, a summary here:

- Better explain choice in sieve ranges
- Use 8 NROYs to quantify uncertainty of final result

Line by line

Abstract (no line numbers) just some word-smithing suggestions, results 'have' implications instead of 'yield' implications. Suggest saying something about how the decreased computation time allows users to model glaciers with ensembles, the way to go to test parametrizations, allows for uncertainty quantification, the list goes on. This goes well beyond modeling glaciers in the Alps better.

Line 7, 'one of the better reconstructed' reword..

Line 12, 'also help reconstruct' reword...

Line 28 – these trimlines are debated all over the world (not just Britain and Scandinavia, but of course you can't cite all)

Line 31 – this paragraph sure has a lot of acronyms. Acronyms are overrated (!), do we really need them at all anymore? Are there space constraints and word counts? Suggest doing away with them, it just makes writing more inaccessible and difficult to understand.

Line 47, this reference of 1-2 km grid should come several sentences earlier when discussing that older model

Line 58, upon reflection, half or more than half of the introduction is just a summary of the results and the paper – that's not really introduction. Yes, many times introductions close with perhaps a final paragraph on 'what we did', but this seems a bit much. Suggest reducing summary text that appears in the introduction, and in place adding more actual intro material. As is, it's like reading results, then going to the results section to read again the same results..

Line 107, not sure I understand the "Are shown; blah" syntax... Change?

Line 143, are you sure "results" is the best description of these first paragraphs – seems more like methods to me. Can you come up with a title that's neither, like "the Instructed Glacier Model" or something?

Line 149, 'follows and improves' well I hope it would improve! It sort of goes without saying, can you replace with 'builds on' ?

Line 156, need word 'materials'?

Line 166, In 'the' main overdeepened valleys...

Line 190 – figure 2/study design... I recommend explaining in the text where the sieve boundaries come from. How are they derived? Why not choose a narrower range? Or a broader range? Is it arbitrary, or based on something? Sorry if I missed it.

Line 203, can you add a comma in that big number (6 digits)

Line 207, can you do away with the footnote and just add the sentence to the paragraph?

Line 212, recommend instead of 'we run a series' write 'we ran 100 simulations from 35 to 18 ka'. I think at this point we know it's using the 300 m grid.

Line 213, data-model agreement is mentioned, but I, for one, didn't understand yet what data – were they introduced yet, or mentioned yet?

Line 226 – how is this range chosen?

Line 233, might be nice here to add that 18 sims are left

Line 244, say '3-step' but that confused me because text describes it as a 2-step. I realize there is a volume step that is derived from ice-thickness (?) so it's kind of like there are two parts of a second step – but I found the description of this as a '3-step' process a little confusing. Maybe wording will help.

Line 246, simulation 37. I agree with your choice to pick one for illustration. I got hung up on this a little bit, because if, in the end, you just pick one among the 100 that matches the data best, then why do the 3-step sieve process? Why have 8 NROYs and not just pick the best? I think you have 8 NROYs (not just one) so you can have some attempt at assigning an uncertainty to your result, no? Maybe you can articulate this better, if that's the case. Otherwise, ignore the sieving and just choose the single best.

Line 274, 'thus permitting to compute' reword..

Line 283, reword for clarity

Line 293, suggest phrasing '... increasing model resolution from x km to x m systematically ...'

Line 297, resulting 'thickness difference' ?
Line 302, replace 'modelled' with 'achieved'
Line 308, remove comma
Line 308/309, huh? Reword?
Line 329, However, we...
Line 331, 'due to too extensive...' reword
Line 402, are these two phenomena not just one?
Line 470, suggest 'cold based regions cover $14\pm 3\%$ of the AIF (NROY mean)(Fig. 6b), more than the 11% modelled at 2 km resolution'
Line 581, change 'positive gradients' to 'increases'
Lines 695 and 890, find a better word/phrase to replace 'hardly' with, which is a bit colloquial.

Jason Briner

(Remarks on code availability)

Reviewer #3

(Remarks to the Author)

(Remarks on code availability)

Reviewer #4

(Remarks to the Author)

In 'A data-consistent model of the last glaciation in the Alps achieved with physics-driven AI' the authors present a much improved simulation of the Pleistocene glaciation of the Alps, which is facilitated through the use of an efficient ice physics emulator. Leveraging the fact that the emulator allows for fast simulations, the authors produce an ensemble of randomly varied simulations. 8 of these produce ice geometries that are in substantially better agreement with observations of ice extent and especially maximum elevation (as deduced from trimlines) than previous studies. Based on the resulting models, the authors draw a number of interesting conclusions about the climatic and glaciological conditions that must have formed the Alpine ice field.

I find this manuscript to be both very interesting and of exceptional clarity. I have no major issues with the work, which I find to be methodologically sound as well as a demonstration of an exciting new hybrid of machine learning and classic glaciological approaches. Indeed, please do not take my relatively few detailed comments (which are below) to indicate that I did not read the manuscript carefully - I read it several times and found that previous points or questions that I had noted were already answered clearly elsewhere in the text. I also believe that this manuscript will be of significant interest to a variety of audiences.

L34: "so-called high-order (three-dimensional)". I'm not sure that any of these terms are relevant to a general audience - these terms (or the subtexts associated with them) are terms-of-art for technical glacier modelers and may not need to appear here.

L37: What does "identical levels of parameter control" mean?

L159: Awkward sentence.

L146--247: I appreciate that Nature-type journals don't want methods in the manuscript, which is why these sections (which clearly are methods) are labeled as 'results'. But maybe it's okay to just call them methods anyways? Or have an 'experimental setup' section before results?

L241: Slightly more detail on what constitutes a 'natural break' and how robust that selection is would be helpful for reproducibility.

L279: What does 'topographically constrained' mean?

L282: It seems that the model remains considerably biased relative to observations, improvement versus previous works notwithstanding. I think it would be of value to try to explain why this remaining bias exists - especially relative to mean ice thickness or something - and if it's something that needs to be considered when contextualizing results.

L408: Why would it be the case that the inclusion of longitudinal stresses produces higher velocities?

L676: What is the numerical method used to update the ice geometry? My understanding is that IGM only emulates the velocity field. If that's not the case, then how is it ensured that essential conservation laws (especially mass) are upheld?

(Remarks on code availability)

I was able to download the code and run the associated simulations. The README file was helpful. I did not run them to completion.

Version 1:

Reviewer comments:

Reviewer #1

(Remarks to the Author)
Review of Leger et al., R1

Many thanks for the thorough rebuttal and the swift resubmission of the manuscript. I have read through the authors response and the revised manuscript and can see the hard work that has gone into doing this. In response to our queries the authors have proposed a new method of comparing the model and the data, which is appropriate and an improvement on the original method. The clarifications requested have all been attended to. There is the requested added detail in the supplementary information and the wider discussion improved. Each comment has had a careful and thoughtful response. I have no further comments for the improvement of this paper and look forward to seeing it published.

(Remarks on code availability)

Reviewer #2

(Remarks to the Author)
The authors did a fine job responding to my comments. I have nothing further.

(Remarks on code availability)

Reviewer #3

(Remarks to the Author)
I co-reviewed this manuscript with one of the reviewers who provided the listed reports. This is part of the Nature Communications initiative to facilitate training in peer review and to provide appropriate recognition for Early Career Researchers who co-review manuscripts.

(Remarks on code availability)

Reviewer #4

(Remarks to the Author)
Thank you for the carefully considered responses to my previous comments. I have no further suggestions.

(Remarks on code availability)

Modifications related to main comments by reviewer 1

#####

Model-data comparison

- We thank the reviewer for raising this very good and important point here on the model-data comparison scheme. Please find below some more information about the original approach, but also regarding a new, more quantitative approach.

- The original model data comparison approach we followed included an initial step (sieve 1) that was semi-quantitative, as based on the visual observation of AIF-wide model-data fit in ice extent with the empirical LGM outline. This was done for every ensemble simulation, as the experiment ran, and was noted down in a spreadsheet. This led to the observation that any maximum modelled AIF areal extent above and below specific capping numbers resulted in clearly too large or too retreated ice extent, and this homogeneously through the AIF (see reply to other reviewer 1 comment regarding why we think that is). The next two sieves of model-data comparison, however, were fully quantitative, using the trimline data with the mean offset, and stdev of offset.

This approach resulted in 8 NROYS that appear to be the best-fitting ensemble simulations, and that are indistinguishable in model-data fit quality in terms of ice extent (when viewing them on a map) whilst minimizing the misfit in ice surface elevation with trimlines. However, we agree that a more quantitative first step would be preferable: especially since it was responsible for a large number of simulations being discarded (82 % originally).

Thus, in response to the comment/question, we designed and tested a different, more quantitative model-data comparison and sieving workflow: presented here:

1) We first rasterize the empirical LGM outline shapefile from Ehlers (and many other studies, see reply to reviewer 1 comment on this) using a polygon buffer wide enough to be rendered in a 300 m resolution grid. Here, after conducting a literature review, we only keep the margins that are described as high confidence due to good moraine preservation and robust dating attributing the margin to around the LGM (anytime between 35 and 18 kyr BP works for us as our model-data comparison is time-independent within that broad LGM timeframe). We do not use the entire LGM outline:

which yields high uncertainty in certain regions (due to lack of precise mapping and dating). We make sure to have data around the full periphery in a broadly homogeneous way, though. This literature review was conducted as part of the ALPICE database project (in prep) by Dr. Sarah Kamleitner.

2) For each ensemble simulation, we isolate the modelled LGM ice margin by using time-independent maximum ice thickness reached for each pixel throughout the full simulation time (e.g. figure 1A). We isolate the ice margin using raster dilation between modelled ice and no-ice masks (using a kernel of 7x7 pixels). This enables to automatically isolate the modelled LGM ice margin as a rasterized buffer zone (see new supplementary Figure 19 and caption).

3) We then add the rasterized empirical LGM outline and the modelled LGM ice margins. We then calculate the total number of pixels that overlap between the two. The higher that number, the better the fit in ice extent for the LGM. We calculate that number for all ensemble simulations, normalize the full list between 0 (worse sim) and 1 (best sim). This gives us a final score between 0 and 1 for each ensemble simulation. This unique number estimates the fit in LGM ice extent across the entire AIF.

If we apply this method and use this score as our first sieve: by only keeping simulations with a score above 0.8, for instance, and then follow with the two other sieves comparing modelled max surface elevation with trimline data, we find that we end up isolating the same NROYS than our 8 original NROYS, using a fully quantitative approach.

However, this is not perfect and we find this also produces two extra outlier simulations besides the 8 NROYS, which are not filtered by this sieving procedure, and which clearly represent a worse fit in ice extent than the 8 indistinguishable NROYS, when their map of maximum time-independent ice thickness is visualized (e.g. Fig. 1a). In these 2 outliers, we find modelled maximum ice extent in the Western and Northwestern AIF (which we know to be well dated and mapped) ends up being far too extensive. We believe this is due to the human eye using many criteria for an assessment of model data fit in ice extent (e.g. overall fit and magnitude of overlap, underlap, with implicit weights), while such a complex assessment is very difficult to fully automatize, and in our fully quantitative method, only one hard criteria is used. Increasing the harshness of the sieves further ends up getting rid of some of the 8 NROYS, whilst keeping 1 or 2 of the outliers. If kept like this, the resulting "new" NROYS (n =10) would become distinguishable in quality: one could clearly see that some simulations within the NROYS are better than others in model-data fit. This would be a bit problematic.

Therefore, we propose this new approach (preferred):

1 - Our new quantitative scoring of ice extent fit (presented above) is used: and a sieve of 0.8 is used (simulations scoring below that are excluded). This is sieve number 1.

2- We fully get rid of the upper and lower cap on maximum total AIF modelled areal extent from the procedure: which was semi-quantitative.

3 - As per the original scheme, the mean and stdev of the misfit on maximum ice surface elevation are used as sieves number 2 and 3. Here, threshold values of 160 m and 100 m are used, respectively, in order to isolate 10 best-fit simulations.

4 - However, we keep a final visual assessment of the 10 remaining simulations to assess whether they are truly indistinguishable in model-data fit quality with regards to ice extent fit the empirical outline.

Due to the small number of simulations left (n=10), this visual check is relatively quick and easy, and is here conducted by 3 different experts within our co-author team.

- From this final check, 2 clear outliers are identified (in consensus) for their worse fit, due to far too extensive ice being modelled along the Western and Northwestern AIF margins, and are thus removed (Fig. 2f).

-We prefer this approach as the human visual observation and expert judgement only comes in the end, and filters out only 2 simulations (2 % of the full ensemble). In the original approach, a semi-qualitative part of the procedure (sieve 1) was also based on visual observations and filtered out 82 % of the ensemble. We thus consider this new method to be more robust, less subjective, and more reproducible. In line with the implementation of this new procedure, we updated the method and results sections, and figure 2 accordingly, and added a supplementary figure with a caption that explains the workflow for scoring model-data fit in ice extent using a quantitative scoring scheme.

To summarize:

Thanks to the reviewer's comment, we modified the workflow and manuscript to apply a new, more quantitative, model-data comparison method for the LGM ice extent analysis. The results and list of 8 NROYs are unchanged. However, in order to achieve the same quality in results as before, we find we still have to apply one visual assessment of the final pool of simulations that passed all quantitative sieves (n=10). This final check results in the removal of 2 outlier simulations. We consider this new approach represents a much less impactful human intervention than before (2% instead of 82%) and is therefore much less subjective. However, if the editor and/or reviewer prefers the original approach, we are of course happy to reverse back to that as well.

#####

Furthermore, in response to this comment: *“The Ehlers limit is fairly dated now and on reading it wasn’t clear to me if this was used as published or if it was updated based on the more recent works cited.”*

We apologise for not making this clear enough, we have now added a sentence after introducing the outline to better explain that the outline used in this study has indeed been updated by numerous studies since Ehlers et al., and thus represents the most up to date level of knowledge (at the AIF scale) on the position of the AIF margins during the LGM:

“Model-data agreement is first evaluated for ice extent by comparing modelled maximum ice thickness fields against an empirical reconstruction of AIF margins at the LGM originally produced by Ehlers et al.¹⁷. In this study, we use the newest version of this outline (Fig. 1a), which was updated by a series of studies^{5,8,47,49-54} providing new constraints on LGM margins of main AIF outlet glaciers (see ‘Methods’ section), and which is used widely within the community^{5,6,8,18,47,48}.”

We have also added an entire paragraph to the methods section on this: called “The empirical LGM outline of the AIF” which tracks the history of updates made to the outline by various studies since. We thank the reviewer as this is a great addition to the methods. However note that adding these references causes us to go over the maximum number of references allowed (n = 70): which we have already been struggling to respect in the original manuscript’s version (we had reached the limit). Would the editor be ok with us going over this threshold? : see the new methods paragraph here:

“The empirical LGM outline of the AIF

The empirical LGM outline of the AIF used in this study was originally produced by an Alps-wide compilation of geomorphological and geochronological evidence by Ehlers et al.¹⁷. Since this key study, the outline has been updated by a series of empirical investigations improving the quality of LGM margin reconstructions in specific sectors of the Alpine foreland. In this work, we use the most up-to-date version of the outline. Here, we provide a summary of the main studies which have updated the LGM AIF outline since Ehlers et al.¹⁷. Gianotti et al.^{50,51} have made updates to LGM margins of the Ivrea outlet glacier in the Dora Baltea region. Braakhekke et al.⁴⁹ have done so for the Orta region, Kamleitner et al.⁵ for the Verbano region, Ravazzi et al.⁵³ for the Oglio region, Monegato et al.⁴⁸ for the Garda region, Federici et al.⁴⁷ for the Gesso region, Ribolini et al.⁸ for the Stura region, and Ivy-Ochs et al.⁵² for the Dora Riparia region.”

#####

Furthermore, in response to this comment: *“The ‘quality controlled’ reference to trimlines in caption of figure 1 – this was not clear to me what this entailed.”*,

We apologise for the confusion and have now added new text to the figure 1 caption guiding the reader to the paragraph in the methods where we explain our quality-check procedure for the trimline data:

“Panel b displays the model-data agreement in Alpine Ice Field ice thickness shown for our best-fit IGM ensemble simulation (number 37), obtained by computing the difference between 353 quality-controlled trimline elevations from literature^{10–16} (see Methods section: “Empirical trimline elevation data”) and maximum modelled ice surface elevations (time-independent).”

#####

Furthermore, in line with reviewer 1’s comments on moraine dating uncertainties: we have added a clearer explanation in our “Model-data comparison scheme” section of the results, regarding the fact that our model-data comparison is time-independent:

“Here, the modelled LGM state for each simulation is obtained by computing the maximum thickness reached in each pixel at any time during the simulation, thus providing a time-independent map of maximum modelled thickness and extent (Fig. 1a). The precise dated timing of the LGM in specific regions is thus not directly compared with the timing of the modelled LGM as part of this model-data comparison scheme, which exclusively aims at finding simulations with the most data-consistent AIF geometry (see Supplementary Fig. 19).”

Therefore, in this work, we do not assume that all moraines are the same age and are well dated. In fact, the moraines described as “LGM” in age by the empirical community, and summarized in the outline, could have been deposited anytime between 35 and 18 kyr BP, the start and end dates of our simulation: since their timing does not impact the results from our simulations and model-data comparison.

However, we fully agree that significant uncertainties lie in the location and confidence of the LGM outline due to spatially heterogeneous quality of geomorphological and geochronological constraints and due to data gaps in understudied regions. In the new scheme, sieve 1 (which automatically scores simulations based on fit with the extent) makes sure to only use the outline in regions of higher confidence due to more data, and based on a literature review (see above). This information has now been added to the caption of the supplementary figure 19 that explains this scheme for scoring.

#####

Furthermore, in response to this comment: *“Trimlines are also challenging to date – often boulders and erratics here will be transported supraglacially and therefore have an inheritance when used in cosmogenic exposure age dating. In the UK for example, trimlines have proven hard to date and age is debated.”*

The trimlines in the Alps used in this study are not dated. Only bedrock transitions from ice-moulded to frost-shattered and weathered have been observed and the few attempts to date them with cosmogenic dating have resulted in very young bedrock ages (Ivy-Ochs, personal communication), due to the very high erosion rates of the high alpine environments and its steep topographies. These very high erosion rates have pushed the community to generally assume that most trimlines in the alps are likely LGM in age, as bedrock evidence of maximum thickness limits from pre-LGM Quaternary glaciations would have been eroded away, and would not be observable today. We agree this assumption yields uncertainties, and we have now added text in line with this to the results section, also in response to a comment on remaining model bias by reviewer 4 (see our reply there). However, the fit we obtain between modelled maximum ice surface elevations and observed trimlines is now very close (only +146 m of offset, and a R2 of 0.92), and we believe this good fit shows that either: 1) the assumption that trimlines are all LGM in age is robust, or 2) if certain observed trimlines were formed by pre-LGM glaciations, then the former maximum ice surface elevations at that time were very similar to those during the LGM.

#####

Wider implications (discussion)

We thank the reviewer for raising these very interesting and relevant points.

We fully agree that the reduced model-data misfits related to this new ability to model at significantly higher resolutions over longer timescales, and at much lower cost than before, thanks to GPU-based models and physics-informed machine learning, should also spark motivation to gather more field data on past glacier and icefield reconstructions to bridge data gaps and better calibrate our models. We fully agree that this could be mentioned more in our discussion.

Indeed, using trimline data in complement to moraine data enables to also obtain a more realistic model in ice thickness, however there are many ways and possible

datasets to better constrain a model from field data, some of which are better adapted to certain regions due to the topographic settings, the nature of the ice cover, and the type of data available to collect, date, and map.

In the Alps, for instance, the high level of topographic constraint on the former icefield makes trimlines a good approach as numerous nunataks existed. This is also the case for many other paleo ice fields (e.g. ice fields in Patagonia, Alaska, New Zealand, Turkey, high mountain Asia and many others). In other regions, with less high topographies and less nunataks, model-data comparison and model hindcasting for paleo reconstructions can be improved through other means, such as calibrating models against ice flow direction using lineations mapping (e.g. Scandinavian ice sheet), using erratic transport and provenance databases (e.g. British-Irish Ice Sheet), detailed mapping and dating of margin retreat patterns and englacial stratigraphic isochrones (e.g. Greenland Ice Sheet), nuclide dipstick models and constraints on past thinning rates (e.g. Antarctica) etc... In this work we don't necessarily want to just focus on a wider implications message that mentions trimlines and nunataks only, as these are one type of constraint amongst many others: but more on the fact that high-resolution modelling will enable to reduce model-data misfits and should thus motivate more data collection more generally.

In line with these thoughts, we have now modified and added text to the relevant paragraph of the discussion:

“The novel approach presented in this study also has implications for improving the reconstruction of other Quaternary ice fields and ice sheet histories. Indeed, achieving spatial resolutions required for a data-consistent LGM model of the AIF was previously unattainable with traditional central-processing-unit-based models. This study demonstrates that physics-driven machine learning and GPU-based processing makes simulating a continental-scale ice field at high (300 m) resolution with a thermo-mechanically coupled, three-dimensional, and high-order model, possible. This is moreover achieved at a fraction of traditional models’ computational costs and with identical parameter modularity. We thus argue that physics-informed, GPU-based models open the door to a new era of more accurate paleo ice field and ice sheet modelling. In particular, the ability to more accurately model the complex dynamics of topographically-constrained large-scale ice fields, such as the AIF during the LGM, through high-order modelling using higher spatial resolutions and wider ensemble-type explorations of parameter spaces, will likely reduce model-data misfits in many other studied mountain ranges. Investigations that would likely benefit from this novel modelling approach include the reconstructions of former large-scale ice fields during Quaternary glaciations in Patagonia, Alaska, high-mountain Asia, Turkey, Georgia, California, Peru, the Pyrenees, New-Zealand, and Iceland, to name only a few. This

improvement in modelling capabilities should also spark new motivation to collect new field data to better constrain the former evolution of ice extent (e.g. with more detailed terminal and lateral moraine mapping and dating⁵), ice thickness and thinning rates (e.g. with new trimlines and cosmogenic nuclide dipstick data⁶⁹), and ice internal flow direction and dynamics (e.g. with lineation mapping and flow set reconstructions⁷⁰, new erratic transport and provenance databases, borehole data and englacial stratigraphic reconstructions^{71,72}) of other former glaciers, ice fields and ice-sheets. This has implications for not only new scientific discoveries in paleoglaciology, but also across other disciplines studying processes linked to Quaternary glacial history, including archaeology, paleoclimatology, paleoecology, and geomorphology.”

#####

Furthermore, in reply to this comment: *“The perturbed ensemble approach to maximise model-data fit will be critical in forward modelling these icefields and demonstrates the importance of hindcasts and data model integration to calibrate projections. This can be emphasised more – there is a rather vague reference to Greenland and antarctica in the conclusion but here, the thicker icesheets have fewer nunataks and it is likely the smaller mountain icefields more similar to the Alps that this approach will be most beneficial. Is the model set up useful for forward modelling of the Alps ?”*

The idea of our last discussion paragraph was to focus on the benefit of our new GPU-based, high resolution and ensemble modelling approach to future projections of cryospheric change. In this final paragraph, we mainly wanted to focus on the Antarctic and Greenland ice sheets for three reasons: 1) The future societal implications, 2) because modelling their present-day states and future evolution is very computationally demanding at high resolution due to their size, and thus there exist a potent computational bottleneck there as well (which is not the case with most other ice fields and mountain glaciers which can already be modelled at high-resolution with traditional high-order models e.g. Elmer Ice, due to the short timescales) , and 3) also because paleo hindcasting of these large continental scale ice sheets has an impact on future projections since they yield considerable memory and inertia from past climate change accumulated over millennial timescales, and which can create multi-century delay in responses to climate change and is thus important for societal implications to better constrain. Our approach thus has implications for the paleo initialisation procedures when modelling the Antarctic and Greenland Ice sheets. Regarding this last argument 3): The memory mechanism is not so important for mountain glaciers and smaller ice fields which do not have enough ice thickness to accumulate such a memory, and for which paleo hindcasting in this way is not really an advantage. Inversion schemes (which are a different type and much larger type of ensemble) of present-day observed

velocity and SMB conditions has been shown to do a good job at representing the state of these contemporary smaller glaciers and icefields (e.g. Millan et al. 2022).

It is true that GPU-based models such as IGM can also provide great improvements to inversion schemes (which are expensive to run) for present-day glaciers and ice fields, and also enable global glacier forward modelling in 3D and high order, whilst the latter has so far been restricted to SIA flowline modelling (e.g. with OGGM). However this is a very different type of modelling to what we do in this study, and has already been talked about in detail in the publication of Cook et al. (2023) who used IGM to inverse and project the future of all glaciers in the Alps, and who is currently working on implementing the same technique to all glaciers and icefields in the world. We feel it would be a bit outside the scope of our study to mention this in our discussion.

However, we believe our approach could really help improve the modelling of the future evolution of major ice sheets (i.e. Greenland and Antarctica). In line with these thoughts and also more minor comments by reviewer 1 regarding the last discussion paragraph, we have tried to modify the text and add new sentences to make this last point less vague and clearer:

“We also believe this new technology may permit more accurate simulations of past and future Greenland- and Antarctic-ice-sheet change, by facilitating ice-sheet-wide simulations at considerably higher resolutions than previously achieved and at lower computational costs. This would allow for broader explorations of parameter spaces and ice-sheet evolution scenarios, with implications for more accurately modelling basal conditions and ice-ocean interactions towards grounding lines, crucial for better projecting the future response of continental ice sheets to climate change. Moreover, reducing Quaternary model-data misfits, as is achieved here with the AIF, may be important for improving the paleo initialisation procedures of Antarctic and Greenland ice sheet models, which greatly impact their future projections. Therefore, we expect physics-informed, GPU-based models to bring advances in projecting future global sea level rise and environmental change associated with the response of contemporary ice sheets to climate change.”

Response to reviewers' line by line comments

#####

Reviewer 1

-Reviewer:

"I wondered how lakes that formed in over deepenings were accounted for and if frontal ablation was simply neglected. Does the model assume that all glaciers are land terminating? Frontal ablation is likely to be a significant driver of ice dynamics so although this is not an issue to be corrected it would warrant a minor sentence or two to clarify."

Authors:

We thank the reviewer for raising this point. At the moment IGM (which is still fairly new) does not have a proglacial lake module (although this is planned as a future point of model development). In this experiment, the model therefore assumes that all glaciers are land terminating. Although not perfect, this assumption is likely to not cause a misrepresentation of the modelled AIF during the LGM (the only event we focus on in this study), since the majority of over-deepened basins susceptible to host a proglacial lake were eventually filled with ice during maximum AIF expansion. The presence of proglacial lakes may slow down the rate of frontal advance during buildup a little (although this would need testing), but the magnitude of glacier advance (which is what we exclusively test here) is not likely to be significantly influenced, we believe. This assumption is furthermore validated by the quality of fit we obtain between our modelled LGM margins and mapped / dated LGM moraines. However, if one wants to model the post-LGM deglaciation of the AIF from those margins and test that event against data, then we believe adding proglacial lakes to the model would be very important, in order to more accurately estimate the rates of retreat. Here, we intentionally stopped our LGM simulations before the major deglaciation of the AIF, which occurred shortly after 18 kyr BP.

We agree this needs to be clarified in the text, and have thus added two sentences to the methods "Surface mass balance" section:

"Note that no proglacial lake module is implemented in this model setup, meaning that all ice is assumed to be land-terminating. We consider this assumption to have little

impact on the LGM geometry of the AIF since most overdeepened basins of the Alpine foreland were eventually ice-filled during maximum glacier advance.”

-Reviewer:

“In the experiment design (line 218) please clarify: are the parameter spaces weighted or selected according to probability or known ranges?”

Authors:

Many thanks for pointing this out, indeed the parameter space is initially defined by given ranges for each of the ensemble-varying parameters (n=10). These ranges, for each parameter, are provided in Supplementary Table 1.

We have now added this to the relevant sentence:

“Within this ensemble, we vary 10 key model parameters drawn from eight different components of the glacier system (see Supplementary Table 1), sampled between given ranges using a Latin hypercube algorithm⁴⁰, optimised with a maximin criterion.”

-Reviewer:

“Line 4 -7: It is helpful to cite some references for mentioned ‘an abundant library of mapped and dated ice-contact glacioenic landforms and sediments separately for reader’s interest.”

Authors:

We fully agree with the reviewer on this point. Unfortunately, we have already reached the limit number of references allowed (n=70) and thus had to make hard choices as to which papers to cite. The study covers many subjects from different fields and thus requires numerous key references making us reach the number of 70 references quickly.

It will be challenging to add new references (although we agree that would be more appropriate). However what we can do is add relevant references that are already being cited later in the manuscript within these first sentences to better guide the reader towards these studies. Please see the change here: 18 relevant references are now part of these first sentences while originally, only 6 were.

“The extent, thickness, and flow geometry of the European Alpine Ice Field (AIF) during Quaternary glaciations has been studied for more than a century¹⁻⁴. The result is an abundant library of mapped and dated ice-contact glaciogenic landforms and sediments (e.g. moraines and tills)⁵⁻⁸, glaciofluvial deposits⁹, and trimlines¹⁰⁻¹⁶, making the extent of the AIF during the Last Glacial Maximum (LGM: ~30-19 ka^{17,18}) one of the most well known in the world^{11,18}.”

-Reviewer:

“Line 15-17: Name the glacier evolution models (for example with some references)”

Authors:

We have added these details to the text. To our knowledge, only PISM and iSOSIA have been used to model the entire AIF during the LGM or other Quaternary maxima. Other studies within the list of references provided (only two others) have used in-house SIA models with no names:

“Under these motivations, over the past 20 years, several studies²⁰⁻²⁴ have produced simulations of AIF glaciations using glacier evolution models such as the Parallel Ice Sheet Model (PISM²⁵) and the integrated second-order shallow ice approximation model (iSOSIA²⁶), with the main objective to closely fit moraine and trimline evidence.”

-Reviewer:

“Line 39: What is the transient time-step of model simulation results, is it the same as the temporal resolution of climate dataset?”

Authors:

Thank you for raising this important question. The answer depends on which components of the model we look at.

-The time step of the iceflow module which updates the ice thickness is flexible depending on minimizing computational cost while respecting the CFL condition and thus is based on maximum ice velocities modelled in previous time step. In our case, during most of the simulation, it is down to 0.001 yr: ~half a day. This is necessary to ensure mass conservation and is the case in most glacier evolution models.

-The time step of the SMB module, i.e. how often the climate fields are updated to compute the annual SMB, using the input climate data (only two snapshots in time: the LGM and the PI), the PDD model, and the glacial index scheme: in our case is updated monthly. The annual SMB is then transmitted to the ice-thickness update once a year (on 1st of October: end of melt season).

-The frequency of saved model outputs for post-processing analyses: this is saved every 50 years in NetCDF files. For the analyses that we conduct, there is little need to produce outputs more frequently than that, bearing in mind that a single simulation output at 50 year intervals is a NetCDF file of 80 GBs in size due to number of variables we need to output and the large size of the grid. The ensemble of 100 simulations (plus previous tests) thus represents a large amount of data to handle during post-processing (~10 TB).

-Reviewer:

“Line 169: (I would expand more on ensemble for clarification: is it ensemble of 100 parameterized model simulations resulting from a group of parameter sets and /or a group of climate input used in the study)”

Authors:

Thank you for raising this point. The 10 chosen parameters lead to simulation-dependent modifications in both ice properties and boundary conditions (including climate and SMB, but also basal conditions, bed topography, etc). More detailed information on this are provided in supplementary table 1, however we have added a few details to the relevant sentence for clarification:

“Within this ensemble, we vary 10 key model parameters drawn from eight different components of the glacier system which modify both ice properties and boundary conditions (see Supplementary Table 1). The 100 parameter sets are sampled between given parameter ranges using a Latin hypercube algorithm⁴⁶, optimised with a maximin criterion.”

-Reviewer:

“how is the initial range for each parameter selected? Is it from the output of the model simulations or based on the model default values or from literatures?”

Authors:

We thank the reviewer for raising this important point.

For most parameters which are transferable between different models, the initial range of parameter values is selected to be a conservative range (in order to explore quite widely the possible model responses) but bracketing values found to best represent the conditions of the AIF during the LGM, as found in the literature and by previous modelling efforts (mostly from previous modelling work of Juergen Mey, Julien Seguinot and Guillaume Jouvét). For ensemble-varying parameters that are specific to IGM and this experiment, however, the initial ranges were obtained by conducting a series of sensitivity tests before running the ensemble and bracketing reference values that generated a better model data fit.

We have added this information as a caption to Supplementary table 1 for increased clarity:

*“*The initial ranges of parameter values that are sampled from (by the Latin Hypercube algorithm) are selected to be a conservative range (thus exploring quite widely the possible model responses) bracketing values found to best represent the conditions of the AIF during the LGM, as found by previous modelling efforts, including the previous work of Mey et al. (2016), Seguinot et al. (2018), Jouvét et al. (2023). For ensemble-varying parameters that are specific to IGM and this experiment (i.e. the catchment-specific precipitation offset parameter, and topographic control on yield stress parameters), initial ranges were obtained by conducting a series of sensitivity tests prior to running the ensemble and targeting ranges that bracket values shown to produce a better model data fit.”*

-Reviewer:

“Line 225 -227: Any explanation for why the maximum AIF areas outside the range of 156-165X 103 produce substantial under- or overestimations of the LGM ice extent reconstruction?”

Authors:

As part of our answer to the main comment of reviewer 1 on the model-data comparison scheme (now more quantitative) and the changes made to the relevant text (see separate response), this statement is no longer part of the paper. However it is very interesting to still address why this happens. We hypothesise this is likely related to LGM

moraines of the Alpine foreland being located on the edges of overdeepenings eroded by multiple quaternary glaciations of similar cooling amplitudes, and which create sweet spots where modelled ice margins can easily stabilise due to reverse bed slopes and voluminous moraine dams. However there likely exist threshold points whereby if the modelled AIF is being pushed enough to grow more, the ice suddenly overflows these moraine dams and is then able to flow much more extensively into the foreland due to down-sloping beds. Contrastingly, if the model is not being forced to grow enough to these sweet spots, then the reverse bed slopes cause most outlet glaciers to be far too retreated. The range of $156\text{-}165 \times 10^3 \text{ km}^2$ is likely to represent the boundaries of these sweet spots: which are thus a consequence of the highly topographically-constrained nature of the AIF during the LGM.

-Reviewer:

“Line 228- 230: Could it be because of the deep learning algorithm that the model is producing similar patterns for outlet glaciers despite extensive changes in the parameters? does this also mean these outlet glaciers control the evolution and behaviour of the entire AIF?”

Authors:

Many thanks for raising this really interesting point. As previously mentioned due to changes made to the model-data comparison scheme (see separate reply to main comments), this statement (which originally came from a semi-quantitative observation) is not part of the paper anymore.

However, we do not believe the iceflow emulator would be responsible for this, as it models ice flow with high fidelity to the Blatter-Pattyn solver (see Jouvét et al. 2023: IGM emulator paper) and is thus expected to produce very similar results to any traditional high-order model. Such a spatially-homogeneous response in modelled AIF-wide extent was already observed in our previous modelling work (as part of the PISM modelling study of Jouvét et al. 2023), and we believe it is related to the highly topographically-constrained nature of the AIF during the LGM. As the ice divides and ice flux patterns are not able to change between simulations due to being heavily controlled by valleys and drainage basins, the different extent and volume response of the AIF are heavily dominated by changes in the surface mass balance controlled by the PDD melt factors and refreezing parameters (see supplementary Table 1), whilst parameters controlling basal conditions (thus sliding) and ice properties have relatively less impact. At the scale of the entire AIF (which is what is shown with these time series in Fig. 2), this leads to parallel time series and spatially-homogeneous responses in modelled AIF extent, as

those climate and PDD parameters are not space-dependent (the change from one simulation to the next in “climate” or “SMB” forcing is the same across the domain). We also agree that the evolution of main outlet glaciers and the rest of the AIF are thus highly coupled.

-Reviewer:

“Line 241-243: how is this natural break for mean and standard deviation i.e. 180 m and 100 m in elevation difference chosen? (is it from sieved simulation data set?)”

Authors:

Originally, it was found by observing sudden jumps in model-data fit in ice extent: where groups of simulations more closely respected the empirical outline when under certain sieve values. However this was semi-quantitative as based on observations as well. We have now changed the model-data comparison procedure to a more quantitative approach (see reply to main comment on this), and now choose the 3 sieve threshold values in order to incrementally remove 90% of the simulations and end up with a final pool of 10 best simulations (10% of the ensemble) which we then conduct a final visual check on. This new approach results in the same 8 NROYs as were found before: as these clearly stand out as the best-fit simulations and are indistinguishable in quality of model-data fit. This protocol has been updated in the text accordingly: in the results section “model-data comparison scheme”:

“Simulations with mean and standard deviation values exceeding 160 m and 100 m in elevation difference, respectively, were sieved out (n=32). This quantitative three-sieve model-data comparison procedure was designed to isolate a group of 10 best-fit simulations (10% of the ensemble) for further inspection. Finally, a visual check of these 10 simulations by three experts enabled to identify two outlier simulations which produce far too extensive ice margins throughout the Western and Northwestern Alps. After removing these two outliers, the final eight Not-Ruled-Out-Yet simulations (NROYs) are found to be indistinguishable in quality and ice-extent fit with the empirical LGM outline and are thus used for all subsequent quantitative results reported in this study, using NROY means and standard deviations in given metrics (Fig. 2).”

-Reviewer:

“Line 247: the best fit to moraines ensemble member selection: was how was this quantified? Visually?”

Authors:

Yes this is a great point. Simulation 37 can be isolated as our best-fit simulation almost entirely quantitatively. If increasing the harshness of our three sieves (i.e. in the new scheme: extent score > 0.84, mean of misfit with trimlines <148 m, stdev of misfit with trimlines <96 m) incrementally and homogeneously across sieves such that the number of remaining simulations is progressively reduced: we basically end up with two last simulations that present the best model-data fit scores: number 30 and 37. Number 30 is one of the outliers which we identified during final visual check: due to far too extensive ice along the western and northern edges of the AIF.: and is thus clearly worse than simulation 37 in model-data fit. Therefore, amongst the 8 NROYs identified (thus after removing the two outliers), the amalgamated model-data fit statistics of simulation 37 are better than the 7 others. However this simulation is only selected for visualization purposes: the 8 NROYs are pretty much indistinguishable in quality of model-data fit (when analysing them visually) and all quantitative results of the paper are thus reported taking into account results from all 8 NROYs (more detail to the text has been added to clarify this, see response to reviewer 2’s comment on this topic).

-Reviewer:

“Line 280 – unprecedented is a very strong word, can it be justified? I would avoid using more superfluous language personally.”

Authors:

We thank the reviewer for pointing this out and agree this sentence sounded a little too dramatic. We have now reworded to:

“Overall, our model produces a thinner LGM ice field resulting in more valley-confined ice flow, and leading to a substantial increase in model-data fit in ice surface elevation.”

-Reviewer:

“Line 326-327: why is that so? Is it because this outlet glacier is influenced by more local phenomena than the Alps-wide phenomenon and the ensemble parameter sets could not capture the processes and drivers?”

Authors:

We thank the reviewer for this highly relevant question. Reviewer 4 asked a similar question related to what we think could explain the remaining model-data biases in both thickness and extent. We have provided three main groups of reasons for what we think to be the most impactful mechanisms in our reply and have added a paragraph to the results section on this: see our reply to reviewer 4 further down. Also: here is the paragraph we added to the main text:

“We believe remaining model-data misfits in ice thickness ($+146 \pm 12$ m) and in the extent of certain outlet glaciers can either be, 1) Reduced further with higher-than-300 m resolution simulations and more extensive parameter-space explorations through larger ensembles, 2) Related to unavoidable uncertainties inherent to paleo trimline and LGM moraine identification due to challenges with landform preservation and dating, and 3) Related to uncertainties and biases in our input climate and surface mass balance parameterisation. A combination of impact from these three mechanisms is also considered likely. For the latter, future improvements in the input climate through higher-resolution regional climate modelling and more transient simulations accompanied by a more complex energy balance model may help reduce some of the remaining model-data misfits.”

-Reviewer:

“Line 408: explain briefly what high-magnitude venturi effects mean (not necessarily all glaciologists know what it means)”

Authors:

Yes thank you for pointing this out, it is true that this term is mostly used in fluid mechanics and in climatology, not so often in glaciology (although very relevant, we believe). We have added this second sentence here:

“When ice in such valleys drains large glacier catchments, NROs produce greater flow convergence, with higher-magnitude venturi effects causing ice flow speedups (Figs. 5, 7). The venturi effect here applies as it describes the increase in an incompressible

fluid's velocity as it passes through a constriction, in order to respect the principle of mass conservation."

-Reviewer:

"Line 557 – how good is the chronological evidence here?"

Authors:

The relevant sentence we wrote in the manuscript is: *"The modelled Rhône outlet glacier reaches maximum extent around 24.6 ka, five centuries after most glaciers (including the nearby Rhein outlet glacier) started to retreat from their maximum positions."*

And here is a citation from the review by Ivy-Ochs (2015) about this specific margin:

"¹⁰Be exposure dates for boulders of hornblende granite brought from the southern valleys of the Valais located along the right lateral position indicate initial glacier withdrawal no later than 24±2 ka (Ivy-Ochs et al., 2004)"

New geochronological constraints were also produced by Hofmann et al. (2024), Graf et al. (2015), and Rey et al. (2017 and 2020).

Here is a citation from Hofmann et al. (2024) regarding outlet glaciers of this region (Rhône, Reuss):

"According to optically stimulated and infrared-stimulated luminescence ages, glacio-fluvial and glacio-lacustrine sediments near the MIS 2 maximum position of the Reuss piedmont glacier (Fig. 5) were deposited at 25.1 ± 2.4 ka and 24.2 ± 2.2 ka, respectively (Gaar et al., 2019). Glacier retreat from the morphologically indistinct moraines at the maximum MIS 2 position of the Reuss glacier, the Untertannwald ice-marginal position, was underway by 21.3 ± 1.0 ka"

Thus, the fit in timing with our modelled LGM in this region seems ok. However please note that our model-data comparison scheme is time-independent (as shown on Fig. 1a) and only assesses the fit in LGM AIF geometry with moraines generally described by the community as "LGM in timing", but that could have been deposited anytime between our simulation's start and end times (35 to 18 kyr BP): which safely brackets the entire LGM period. As this is not our objective, in this work, we do not assess the fit in the precise timing of our modelled LGM with dates attributed to LGM margins in each region. We have included more details relevant to this in our reply to the main comment by

reviewer 1 on model-data comparison: and have also made sure to add this to the model-data comparison scheme for more clarity:

“Here, the modelled LGM state for each simulation is obtained by computing the maximum thickness reached in each pixel at any time during the simulation, thus providing a time-independent map of maximum modelled thickness and extent (Fig. 1a). The precise dated timing of the LGM in specific regions is thus not directly compared with the timing of the modelled LGM as part of this model-data comparison scheme, which exclusively aims at finding simulations with the most data-consistent AIF geometry (see Supplementary Fig. 19).”

-Reviewer:

“Line 637 – how else could it inform future investigations – see comments above.”

Authors:

We thank the reviewer for this comment. Please see our reply to the main reviewer 1 comment on the wider implications section of the discussion: to which we have now modified and added more text in line with the reviewer’s suggestions.

-Reviewer:

“Line 647 – there are also implications for data as well”

Authors:

We strongly agree with this, please see our reply to the main reviewer 1 comment on the wider implications section of the discussion.

-Reviewer:

“Line 656 – this is very vague and I suggest rewriting this paragraph. Greenland & WAIS/EAIS have very few nunataks and requires a different approach. Can a very small number of spatially limited dispersed nunataks improve modelling here?”

Authors:

We apologise that these few sentences were not very clear. Originally, we were here referring much more broadly (moving away from the Alps and model-data comparison with trimlines, nunataks etc) to the new technology of GPU-based glacier evolution models and physics-informed machine learning (which is here applied for the first time to a continental scale ice body), and which permits significant gains in computational means and thus in model resolution. What we referred to in this sentence is that modelling the past and future of the Greenland Ice Sheets and Antarctic Ice Sheets, besides having huge future implications for society, is currently limited by computational costs and thus model resolutions (especially in paleo timescale). The Antarctic Ice Sheet, for instance, in paleo timescales is usually modelled at 15 km resolution at best (in the interior). At the 15 km, the interactions between the ice sheet and its bed are very likely to be mis-represented. We know that for these ice sheets, calibrating the models over paleo millennial timescales is important and impacts future projections. Further developing GPU-based models may bring significant improvements in that regard, enabling to model the entire ice sheets at sub-km resolution with homogeneous grids (we anticipate this will soon be possible thanks to the new generation of GPUs).

To increase the clarity of that message, we have reworded and increased the length of that particular paragraph to:

“We also believe this new technology may permit more accurate simulations of past and future Greenland- and Antarctic-ice-sheet change, by facilitating ice-sheet-wide simulations at considerably higher resolutions than previously achieved and at lower computational costs. This would allow for broader explorations of parameter spaces and ice-sheet evolution scenarios, with implications for more accurately modelling basal conditions and ice-ocean interactions towards grounding lines, crucial for better projecting the future response of continental ice sheets to climate change³⁸. Moreover, reducing Quaternary model-data misfits, as is achieved here with the AIF, may be important for improving the paleo initialisation procedures of Antarctic and Greenland ice sheet models, which greatly impact their future projections^{38,41}. Therefore, we expect physics-informed, GPU-based models to bring advances in projecting future global sea level rise and environmental change associated with the response of contemporary ice sheets to climate change.”

Please also refer to our reply to the main reviewer 1 comment on the wider implications' part of the discussion.

-Reviewer:

“Line 659 – a rather superficial sentence to end on. You can finish with a more exciting call to arms and set the agenda for the community here.”

Authors:

We thank the reviewer for this comment and have added a new conclusion sentence that attempts to do more agenda-setting for the community:

“To conclude, we believe the development and application of physics-driven AI and GPU-based glacier and ice-sheet models should become a primary avenue of future research due to its wide and multidisciplinary implications for the fields of Quaternary and cryosphere science.”

#####

Reviewer 2

-Reviewer:

“Abstract (no line numbers) just some word-smithing suggestions, results ‘have’ implications instead of ‘yield’ implications. Suggest saying something about how the decreased computation time allows users to model glaciers with ensembles, the way to go to test parametrizations, allows for uncertainty quantification, the list goes on. This goes well beyond modeling glaciers in the Alps better.”

Authors:

We thank the reviewer for this point. We have rephrased to ‘have’ implications, and have added some more details to the last sentence of the abstract:

“Furthermore, this study demonstrates that physics-informed AI-driven glacier evolution models can overcome the bottleneck of high-resolution continental-scale modelling, and enable to better explore parameter spaces through reduced computational costs, both of which are required by glacier and ice-sheet modelling to accurately describe complex topographies, ice dynamics, and test model parameterisations.”

-Reviewer:

“Line 7, ‘one of the better reconstructed’ reword..”

Authors:

We rephrased this to: *“making the extent of the AIF during the Last Glacial Maximum (LGM: ~30-19 ka^{5,6}) one of the most well known in the world”*

-Reviewer:

“Line 12, ‘also help reconstruct’ reword...”

Authors:

This was rephrased to: *“has implications for understanding such mechanisms and can also improve reconstructions of late-Quaternary human and ecological history”*.

-Reviewer:

“Line 28 – these trimlines are debated all over the world (not just Britain and Scandinavia, but of course you can’t cite all)”

Authors:

We thank the reviewer for this point, which we fully agree with. Of course we are limited in the number of reference which we have already reached the limit of. But here we have rephrased to clearly state that Scandinavia and Britain are only two examples:

“as was previously debated in other glaciated regions, such as Britain and Scandinavia for instance²³.”

-Reviewer:

“Line 31 – this paragraph sure has a lot of acronyms. Acronyms are overrated (!), do we really need them at all anymore? Are there space constraints and word counts? Suggest doing away with them, it just makes writing more inaccessible and difficult to understand.”

Authors:

We thank the reviewer and we fully agree with this comment, it would be nice to move away from acronyms as much as possible. However here we are also limited by word count: with a maximum of 5000 words for the main text and 3000 words for the Methods. As of now, the main text is at 4800 words, so a bit tight already. The IGM, AIF, and LGM acronyms are used more than 50 times each across the manuscript and would thus increase the word count far beyond the limit if spelled out.

The acronym GPU is actually very important to keep (unfortunately). It is the key to the entire new modelling approach followed in this study. Being able to compute on the GPU is what makes the computational gains possible. We believe this term will become

increasingly important within environmental sciences in general and people need to therefore know what it means. Unfortunately, computer scientists only refer to GPUs using the acronym, and never the full term “Graphics processing unit”.

We have however removed the acronym ‘GPU’ from the paper, which was not used enough to be relevant.

-Reviewer:

Line 47, this reference of 1-2 km grid should come several sentences earlier when discussing that older model

Authors:

There is in fact a reference to this resolution earlier in the introduction (originally line 23): with this sentence:

“This ice thickness overestimation has been attributed to oversimplified ice flow dynamics and the coarse spatial resolutions of previous AIF-wide models, computationally restricted to 1-2 km”.

However, the term “previous” was not there and we here added it following the reviewer’s comment to make it clearer that these were indeed the best-achieved resolutions for previous models of the Alpine Ice Field before our study.

-Reviewer:

Line 58, upon reflection, half or more than half of the introduction is just a summary of the results and the paper – that’s not really introduction. Yes, many times introductions close with perhaps a final paragraph on ‘what we did’, but this seems a bit much. Suggest reducing summary text that appears in the introduction, and in place adding more actual intro material. As is, it’s like reading results, then going to the results section to read again the same results..

Authors:

We thank the reviewer for this comment. Indeed the introduction is structured in three paragraphs. The first two describe the context of AIF reconstruction, and the issues with previous modelling efforts and the model-data misfits, respectively (thus highlighting the context and motivation for our work). The third and last introduction paragraph,

however, was written as a summary of the new approach followed in our work, and the key results it produces. We agree with the reviewer this third paragraph could be made shorter and include less results, as it needs to make the reader want to read the rest of the paper without giving away too much detailed information.

We have therefore removed a few sentences from the introduction paragraph 3 regarding specific results. In the new text, results are now only summarized in two sentences (with no specific numbers given, only trends). The rest of the third paragraph is more focused on introducing the nature of the novel method employed (GPU-based modelling) and its wider implications. The latter has been expanded a bit in line with new changes to the discussion following comments by reviewer 1 :

“To address this conundrum, we apply a novel approach by modelling the AIF with the Instructed Glacier Model (IGM)^{31,32}, a thermo-mechanically coupled three-dimensional glacier evolution model. IGM makes use of recent improvements in physics-informed machine learning to accelerate the high-order “Blatter-Pattyn” solver of ice flow³³, enabling efficient computation on graphics processing units (GPU). The resulting model simulates glacier motion with fidelity while offering the same ability to perturbate parameters controlling ice properties as traditional central-processing-unit-based models, but at a fraction of the computational cost. Here, we use IGM to simulate the AIF’s transient evolution from 35 to 18 ka, thus bracketing the full LGM period, at a spatial resolution of 300 m, an order of magnitude higher than previously achieved²⁰. Computationally unfeasible with traditional glacier models, our perturbed parameter ensemble experiment shows that increasing spatial resolution substantially reduces model-data misfits in both LGM ice extent and thickness across the Alps, resulting in a thinner ice field than previous 1-2 km resolution models^{20,22}. This new approach enables us, for the first time, to produce a high-resolution, data-consistent, yet physics-based transient simulation of the AIF’s last glaciation. The results have wider implications for addressing open research questions on Quaternary glacial erosion processes, postglacial isostatic rebound, and paleoclimate in the European Alps. Our study also reveals that GPU-based glacier evolution modelling is a promising tool for better reconstructing other paleo ice fields, but also for past and future ice-sheet-scale modelling. Indeed, by substantially reducing computational costs, this new approach permits advances in modelling resolution and parameter space exploration, both essential to resolve complex topographies and model more accurate past and future glaciated environments.”

-Reviewer:

“Line 107, not sure I understand the “Are shown; blah” syntax... Change?”

Authors:

Many thanks for pointing this out, we have now changed the wording to:

“Panel a shows the 2-D field of modelled time-independent maximum ice thickness for simulation 37”

-Reviewer:

“Line 143, are you sure “results” is the best description of these first paragraphs – seems more like methods to me. Can you come up with a title that’s neither, like “the Instructed Glacier Model” or something?”

Authors:

We here provide the same reply as for a comment by reviewer 4 on a very similar point.

Yes we agree this can be a bit difficult. As far as we understand the main text of the paper should give all the tools for the readers to understand what the experiment is, how it was conducted, the results it produced and their wider implications. More precise details on the detailed techniques employed should be moved to the methods.

However, some methods are simply too unavoidable to not be mentioned in the main text, otherwise the experiment cannot be understood. This is the case with this study which relies on a new approach and new model. This means that parts of the ‘results’ end up being more methods-like. This is why we used explicit headings in this case: with paragraphs being called “High resolution model setup” and “Model-data comparison scheme”. This slight difference with more conventional papers occurs in most Nature Communications papers, we find. The editorial guidelines on how to name different “main” paper sections is fairly strict, it seems, and we don’t believe it is possible to move text between the “introduction” and “results” sections in Nature Communication papers, unless the editor says otherwise and allows it to occur in this case?

We thus let the editor make a decision on this, based on whether it is possible to change this in Nature Communications papers or not.

-Reviewer:

“Line 149, ‘follows and improves’ well I hope it would improve! It sort of goes without saying, can you replace with ‘builds on’ ?”

Authors:

We agree that “*builds on*” would be more appropriate and have now reworded the sentence using that instead, many thanks for pointing this out.:

“we design a high-resolution (300 m) model setup with IGM^{24,25} that builds on the Parallel Ice Sheet Model (PISM²⁷) experimental setup of Jouvét et al.¹⁰.”

-Reviewer:

“Line 156, need word ‘materials’?”

Authors:

Indeed this word is not necessary and has now been removed from the sentence.

-Reviewer:

“Line 166, In ‘the’ main overdeepened valleys...”

Authors:

Thanks for spotting this, we added the word accordingly.

-Reviewer:

“Line 190 – figure 2/study design... I recommend explaining in the text where the sieve boundaries come from. How are they derived? Why not choose a narrower range? Or a broader range? Is it arbitrary, or based on something? Sorry if I missed it.”

Authors:

Yes this is a very good and important point. Thank you. Please refer to our longer reply to reviewer 1 on the model-data comparison scheme, which has now been modified to become more quantitative and robust, we believe. This new scheme moreover produces the same set of 8 NROYs which are found to be indistinguishable in model-data fit quality, which is re-assuring as it shows that slightly different methods in model-data tests (and different sieves) lead to similar results. Increasing the harshness of the sieves further would result in the removal of simulations which are (visually) indistinguishable in their fit with the empirical LGM outline, while their fit with the trimlines are also very similar: and would thus be difficult to justify.

-Reviewer:

“Line 203, can you add a comma in that big number (6 digits)”

Authors:

We have changed this accordingly.

-Reviewer:

“Line 207, can you do away with the footnote and just add the sentence to the paragraph?”

Authors:

We thank the reviewer for pointing this out and have now added the sentence and removed the footnote accordingly:

“While running a 20 kyr-long simulation at 300 m would require ~2.5 years with PISM using a 32-core (3.7 GHz) computer (or ~6 months using a state-of-the-art high-performance computer), it is achieved in only ~2.5 days with IGM using a single GPU (Nvidia RTX 4090).”

-Reviewer:

“Line 212, recommend instead of ‘we run a series’ write ‘we ran 100 simulations from 35 to 18 ka’. I think at this point we know it’s using the 300 m grid.”

Authors:

This sentence was changed accordingly.

-Reviewer:

“Line 213, data-model agreement is mentioned, but I, for one, didn’t understand yet what data – were they introduced yet, or mentioned yet?”

Authors:

Many thanks for pointing this out: the model-data comparison aspect is presented for the first time in the next paragraph, which follows 3 sentences later. We thus added “see below” behind “model-data agreement” in a bracket within this sentence.

-Reviewer:

“Line 226 – how is this range chosen?”

Authors:

Thank for for raising this very good point. However due to the new change made to the model-data comparison scheme (see detailed reply to reviewer 1’s main comment), with a first sieve modified to be a fully quantitative assessment and scoring of model-data fit in LGM ice extent, we now do not have to specify a range for this anymore, and this sentence has now been removed.

-Reviewer:

“Line 233, might be nice here to add that 18 sims are left”

Authors:

In the new text that describes the new model-data comparison scheme, we have made sure to make this more explicit indeed:

“This quantitative three-sieve model-data comparison procedure was designed to isolate a group of 10 best-fit simulations (10% of the ensemble) for further inspection.”

-Reviewer:

“Line 244, say ‘3-step’ but that confused me because text describes it as a 2-step. I realize there is a volume step that is derived from ice-thickness (?) so it’s kind of like there are two parts of a second step – but I found the description of this as a ‘3-step’ process a little confusing. Maybe wording will help.”

Authors:

We agree this was worded in a confusing way before. In the new relevant sentence shown directly above in reply to the previous comment: please note that we have reworded using “three-sieve” rather than “three-step”, which is more explicit.

-Reviewer:

“Line 246, simulation 37. I agree with your choice to pick one for illustration. I got hung up on this a little bit, because if, in the end, you just pick one among the 100 that matches the data best, then why do the 3-step sieve process? Why have 8 NROYs and not just pick the best? I think you have 8 NROYs (not just one) so you can have some attempt at assigning an uncertainty to your result, no? Maybe you can articulate this better, if that’s the case. Otherwise, ignore the sieving and just choose the single best.”

Authors:

Yes indeed we use the 8 indistinguishable NROYs to use for better uncertainty quantification and to find potential patterns (or not) in most suitable parameter values, but also to not be biased towards one parameter configuration. This is why, in the “Improved LGM model-data fit” and throughout the discussion section, we systematically report our results using NROY-mean and NROY-stdev statistics (with ctrl + f you will find the term “NROY mean” comes up more than 10 times throughout the text and captions). However we agree this was not made clear enough in this “Model-data comparison scheme” section.

For more clarity, we have thus added the sentence:

“the final eight Not-Ruled-Out-Yet simulations (NROYs) are found to be indistinguishable in quality and ice-extent fit with the empirical LGM outline and are thus used for all subsequent quantitative results reported in this study, using NROY means and standard deviations in given metrics”

-Reviewer:

“Line 274, ‘thus permitting to compute’ reword..”

Authors:

We have rephrased to: “Note panel **a** only displays coloured pixels where ice is modelled by both IGM and PISM simulations (when maximum ice thickness is reached in each pixel), as this is required to compute the thickness difference”.

-Reviewer:

“Line 283, reword for clarity”

Authors:

We agree this was a little confusing. We have reworded this sentence to:

“The positive bias in modelled ice thickness is reduced by 200% and 450% compared with these earlier studies, respectively.”

-Reviewer:

“Line 293, suggest phrasing ‘... increasing model resolution from x km to x m systematically ...”

Authors:

This has now been changed accordingly.

-Reviewer:

Line 297, resulting 'thickness difference' ?

Authors:

We have now replaced "thinning pattern" by "thickness difference" as suggested.

-Reviewer:

Line 302, replace 'modelled' with 'achieved'

Authors:

We have now replaced it accordingly, many thanks for pointing this out.

-Reviewer:

Line 308, remove comma

Authors:

Nicely spotted, many thanks. Now removed.

-Reviewer:

Line 308/309, huh? Reword?

Authors:

Yes indeed this sentence was a little confusing. We have reworded to:

“Across our 100 simulations, we find highly variable modelled basal ice temperatures at the locations of trimlines during the LGM, with no apparent clusters towards distinct temperatures. This suggests no relationship exists in our modelling between thermal boundaries of basal ice and trimline formation (see Supplementary Fig. 4).”

-Reviewer:

“Line 329, However, we...”

Authors:

Change made accordingly.

-Reviewer:

“Line 331, ‘due to too extensive...’ reword”

Authors:

We have rephrased to: *“However, we note three instances, i.e. the Rhône, Dora-Baltea and Isère outlet glaciers, where our NROs model too extensive ice margins leading to a worse fit with data than Jouvét et al¹⁰’s model (Fig. 4a)”*

-Reviewer:

“Line 402, are these two phenomena not just one?”

Authors:

Many thanks, we now realise this sentence was not 100% clear. By “these two phenomena”, we meant the higher resolution modelling and the reduction in ice thickness. For more clarity, we rephrased to:

“Our higher-resolution model simulations (300 m instead of 1-2 km in previous studies^{8,10}) result in thinner ice and glacier surface lowering across the AIF during the

LGM. We hypothesise that localised ice flow speedup is the main underlying mechanism responsible for this result.”

-Reviewer:

“Line 470, suggest ‘cold based regions cover 14±3% of the AIF (NROY mean)(Fig. 6b), more than the 11% modelled at 2 km resolution”

Authors:

We have made this change accordingly.

-Reviewer:

“Line 581, change ‘positive gradients’ to ‘increases”

Authors:

We have made this change accordingly.

-Reviewer:

“Lines 695 and 890, find a better word/phrase to replace ‘hardly’ with, which is a bit colloquial.”

Authors:

Many thanks for spotting this,

We have now reworded to : *“with negligible accuracy losses”*

And to: *“The impact of removing these 43 presumed outliers on the mean model-data misfit between modelled ice surface elevations and reported trimline elevations is negligible (on average a ~2% reduction)”*

#####

Reviewer 4

-Reviewer:

“L34: "so-called high-order (three-dimensional)". I'm not sure that any of these terms are relevant to a general audience - these terms (or the subtexts associated with them) are terms-of-art for technical glacier modelers and may not need to appear here.”

Authors:

We agree that the term “high-order” is most likely too technical for a general audience and thus removed it from the paper Abstract. In the introduction, we also simplified the sentence “*to accelerate the so-called high-order (three-dimensional) “Blatter-Pattyn” solver of ice flow*”, to replace it by “*to accelerate the high-order “Blatter-Pattyn” solver of ice flow*”. We however consider it important to mention that IGM is informed by high-order physics, and not the SIA or a hybrid (SIA/SSA) scheme, as this matters for obtaining better results in complex topographies, as mentioned in the results and discussion sections. Moreover, we believe most glaciologists (including non-modelers) are now exposed enough to modelling studies to know the difference between SIA, high-order, and full-stokes ice flow physics.

-Reviewer:

“L37: What does "identical levels of parameter control" mean?”

Authors:

We thank the reviewer for pointing this out, this sentence is indeed not very clear. Here we simply mean to say that IGM, despite being a slightly different model than traditional numerical glacier models (CPU-based) due to the ice flow solver being emulated by a neural network, still provides the same ability to play with the same parameters of ice

dynamics and ice flow motion as other models. In other terms, one can tweak exactly the same knobs as before.

To make this clearer, we modified the relevant sentence to:

“The resulting model simulates glacier motion with fidelity while offering the same ability to modify parameters controlling ice properties, but at a fraction of the computational cost of traditional central processing unit (CPU)-based models.”

-Reviewer:

“L159: Awkward sentence.”

Authors:

We thank the reviewer for pointing this out. For more clarity, we reworded and modified the sentence to:

“The experiment setup of Juvet et al.¹⁰ is however improved by using an Alps-specific climate record (instead of the EPICA³⁴ ice core record) as input to our glacial index scheme^{10,35} (Fig. 2e). The applied signal combines the Bergsee lacustrine record³⁶ (35-30 ka: Black Forest, Southern Germany) and the Sieben Hengste speleothem $\delta^{18}O$ record³⁷ (30-18 ka: Bernese Alps, Switzerland).”

-Reviewer:

“L146--247: I appreciate that Nature-type journals don't want methods in the manuscript, which is why these sections (which clearly are methods) are labeled as `results'. But maybe it's okay to just call them methods anyways? Or have an `experimental setup' section before results?”

Authors:

Yes we agree this can be a bit difficult. As far as we understand the main text of the paper should give all the tools for the readers to understand what the experiment is, how it was conducted, the results it produced and their wider implications. More precise details on the detailed techniques employed should be moved to the methods.

However, some methods are simply too unavoidable to not be mentioned in the main text, otherwise the experiment cannot be understood. This is the case with this study

which relies on a new approach and new model. This means that parts of the ‘results’ end up being more methods-like. This is why we used explicit headings in this case: with paragraphs being called “*High resolution model setup*” and “*Model-data comparison scheme*”. This slight difference with more conventional papers occurs in most Nature Communications papers, we find. The editorial guidelines on how to name different “main” paper sections is fairly strict, it seems, and we don’t believe it is possible to move text between the “Introduction” and “Results” sections in Nature Communication papers, unless the editor says otherwise and allows it to occur in this case?

-Reviewer:

“L241: Slightly more detail on what constitutes a ‘natural break’ and how robust that selection is would be helpful for reproducibility.”

Authors:

We thank the reviewer for pointing this out: this section of text has now been updated and modified as the model-data comparison and sieving scheme has changed, in response to reviewer 1’s main comment (see more detailed reply there). We do not aim to identify these natural breaks anymore but instead aim to isolate NROYs that are indistinguishable in quality of model-data ice extent fit with the empirical LGM outline. These natural breaks are not mentioned anymore in the text. The new scheme is more quantitative and thus more reproducible than the original approach, we believe. (see our reply to main comment by reviewer 1 on this).

-Reviewer:

“L279: What does ‘topographically constrained’ mean?”

Authors:

With this term we here meant that the modelled AIF, because thinner than before, is now more coupled to the topography (glacier catchments = fluvial catchments) thus behaving like an ice field as opposed to an ice sheet that is thick enough to decouple from the topography. Indeed, this may not be well understood if termed “topographically constrained”. We thus reworded this sentence to:

“Overall, our model produces a thinner LGM ice field resulting in more valley-confined ice flow, and leading to a substantial increase in model-data fit in ice surface elevation.”

-Reviewer:

“L282: It seems that the model remains considerably biased relative to observations, improvement versus previous works notwithstanding. I think it would be of value to try to explain why this remaining bias exists - especially relative to mean ice thickness or something - and if it's something that needs to be considered when contextualizing results.”

Authors:

Indeed some (although small relative to previous studies) remaining bias in the extent of certain outlet glaciers and of +146 m in ice thickness between our best-fit models and empirical reconstructions can be observed. We believe it can mostly be attributed to three possible mechanisms, or a combination of them.

-1) the remaining ice thickness bias will likely be further reduced if running the simulations at even higher spatial resolution than 300 m and if running larger ensembles enabling a deeper parameter space exploration. At the moment we are unable to run at higher resolution due to limited GPU memory, but this will shortly be made possible by new generations of GPUs (such as the NVIDIA H100) which will undoubtedly permit to model the entire AIF at 100 m and possibly to run larger ensembles. We don't expect this thickness offset to disappear, but we don't know until we can try. The full complexity of the bed topography, especially over steep slopes, is still not perfectly well represented using a 300 m resolution grid. Significantly more realistic ice flow, but also avalanche and snow redistribution processes over steep slopes and rock cliffs would be obtained with a 100 or 50 m topography, which may help reduce the ice thickness offset further in some cases.

-2) Observing trimlines from bedrock surfaces transitioning from ice moulded erosion regimes to weathered and frost-shattered erosion regimes can be challenging and yields uncertainties, especially due to post deglaciation processes that can disturb, erode, and remove the highest elevated ice-moulded bedrock surfaces. The fact that the remaining misfit is only of +~150 m and the R^2 of ~0.92 (Fig. 1b) strongly suggests the assumption that trimlines are formed near the ice surface is valid, in the European Alps at least. However it does not mean a trimline observation should necessarily be considered the exact ice surface. What we mean here is, not all uncertainties are necessarily linked to

the modelling, and it is possible a remaining misfit of ~100 m is unavoidable due to nature of the trimline observation data (for paleo reconstructions) and may represent an acceptable range of uncertainties in this type of model-data comparison. This is just a hypothesis, however, time and further investigations will help validate or refute this.

3) Although the input climate used in this study is likely the best available input climate data for the LGM in the European Alps (Russo et al., 2024), it still only comes from a single regional climate model (WRF), and is just one snapshot in time (the LGM at 24 kyr BP or the PI), and is still 2 km resolution, which may not be enough to capture the full inter-valley variations in past climate and mountain-specific climatology. Future improvements in the input climate through more regional climate modelling reconstructions with more transient climate simulations and at higher spatial resolution will most likely help reduce some of the remaining biases in ice thickness. Once the largest uncertainties linked to regional climate modelling of the LGM are reduced, a better SMB model than the PDD (e.g. energy balance models) could also help better represent some of the missing surface ablation mechanisms (e.g. sublimation).

Based on these thoughts, we have added a new paragraph to the relevant section of the results (end of “*Improved LGM model-data fit*” section) to contextualize the results a bit more:

“We believe remaining model-data misfits in ice thickness ($+146 \pm 12$ m) and in the extent of certain outlet glaciers can either be, 1) Reduced further with higher-than-300 m resolution simulations and more extensive parameter-space explorations through larger ensembles, 2) Related to unavoidable uncertainties inherent to paleo trimline and LGM moraine identification due to challenges with landform preservation and dating, and 3) Related to uncertainties and biases in our input climate and surface mass balance parameterisation. A combination of impact from these three mechanisms is also considered likely. For the latter, future improvements in the input climate through higher-resolution regional climate modelling and more transient simulations accompanied by a more complex energy balance model may help reduce some of the remaining model-data misfits.”

-Reviewer:

“L408: Why would it be the case that the inclusion of longitudinal stresses produces higher velocities?”

Authors:

Yes indeed we see where the confusion could lie here. In this sentence, we meant more that due to Glen's flow law, a high-order scheme such as the "Blatter-Pattyn" model which includes three-dimensional coupling of both the vertical and horizontal components of strain will necessarily lead to a lower ice viscosity under a given stress than when only considering the vertical (SIA) or horizontal (SSA) components of strain separately.

We have rephrased this to:

"Here, the high-order Blatter-Pattyn solver²⁶ used in IGM²⁵ considers both the vertical and horizontal components of strain through the ice column⁴⁹, which helps produce strong ice velocity gradients and promotes the formation of lateral shear margins in valley-confined settings, thus encouraging higher glacier velocities when increasing spatial resolution⁵⁰."

-Reviewer:

"L676: What is the numerical method used to update the ice geometry? My understanding is that IGM only emulates the velocity field. If that's not the case, then how is it ensured that essential conservation laws (especially mass) are upheld?"

Authors:

We thank the reviewer for pointing this out. It seems we forgot to write a key sentence in this part of the method which explains this. It is indeed true to say that IGM only emulates the ice velocity field.

We've now added this important sentence to this paragraph:

"Equation (1) is solved using an explicit upwind finite volume scheme on a regular grid which allows the model to update the ice thickness while conserving mass."

#####